# On Sequential Bayesian Inference for Continual Learning

## Abstract

Sequential Bayesian inference can be used for *continual learning* to prevent catastrophic forgetting of past tasks and provide an informative prior when learning new tasks. We revisit sequential Bayesian inference and assess whether using the previous task's posterior as a prior for a new task can prevent catastrophic forgetting in Bayesian neural networks. Our first contribution is to perform sequential Bayesian inference using Hamiltonian Monte Carlo. We propagate the posterior as a prior for new tasks by approximating the posterior via fitting a density estimator on Hamiltonian Monte Carlo samples. We find that this approach fails to prevent catastrophic forgetting demonstrating the difficulty in performing sequential Bayesian inference in neural networks. Furthermore, we study simple analytical examples of sequential Bayesian inference and CL and highlight the issue of model misspecification which can lead to sub-optimal continual learning performance despite exact inference. Furthermore, we discuss how task data imbalances can cause forgetting. From these limitations, we argue that we need probabilistic models of the continual learning generative process rather than relying on sequential Bayesian inference over Bayesian neural network weights. Our final contribution is to propose a simple baseline called *Prototypical Bayesian Continual Learning*, which is competitive with the best performing Bayesian continual learning methods on class incremental continual learning computer vision benchmarks.

## 1 Introduction

The goal of continual learning (CL) is to find a predictor that learns to solve a sequence of new tasks without losing the ability to solve previously learned tasks. One key challenge of CL with neural networks (NNs) is that model parameters from previously learned tasks are "overwritten" during gradient-based learning of new tasks, which leads to *catastrophic forgetting* of previously learned abilities (McCloskey & Cohen, 1989; French, 1999). One approach to CL hinges on using recursive applications of Bayes' Theorem; using the weight posterior in a Bayesian neural network (BNN) as the prior for a new task (Kirkpatrick et al., 2017). However, obtaining a full posterior over NN weights is computationally demanding and we often need to resort to approximations, such as the Laplace method (MacKay, 1992) or variational inference (Graves, 2011; Blundell et al., 2015) to obtain a neural network weight posterior.

When performing Bayesian CL, sequential Bayesian inference is performed with an approximate BNN posterior, not the true posterior (Schwarz et al., 2018; Ritter et al., 2018; Nguyen et al., 2018; Ebrahimi et al., 2019; Kessler et al., 2021; Loo et al., 2020). If we consider the performance of sequential Bayesian inference with a variational approximation over a BNN weight posterior then we barely observe an improvement over simply learning new tasks with stochastic gradient descent (SGD). We will develop this statement further in Section 2.2. So if we had access to the true BNN weight posterior, would this be enough to prevent forgetting by sequential Bayesian inference?

Our contributions in this paper are to revisit Bayesian CL. 1) Experimentally, we perform sequential Bayesian inference using the true Bayesian NN weight posterior. We do this by using the gold standard of Bayesian inference methods, Hamiltonian Monte Carlo (HMC) (Neal et al., 2011). We use density estimation over HMC samples and use this approximate posterior density as a prior for the next task within the HMC sampling process. Surprisingly our HMC method for CL yields no noticeable benefits over an approximate inference method (VCL Nguyen et al. (2018)) despite using samples from the true posterior. 2) As a result

we consider a simple analytical example and highlight that exact inference with a misspecified model can still cause forgetting. 3) We show mathematically that under certain assumptions task data imbalances will cause forgetting in Bayesian NNs. 4) We propose a new probabilistic model for CL and show that by explicitly modeling the generative process of the data, we can achieve good performance, avoiding the need to rely on recursive Bayesian inference over NN weights to prevent forgetting. Our proposed model, *Prototypical Bayesian Continual Learning* (ProtoCL), is conceptually simple, scalable, and competitive with state of the art Bayesian CL methods in the class-incremental learning setting.

## 2 Background

### 2.1 The Continual Learning Problem

*Continual learning* (CL) is a learning setting whereby a model must learn to make predictions over a set of tasks sequentially while maintaining performance across all previously learned tasks. In CL, the model is sequentially shown $T$ tasks, denoted $\mathcal{T}_t$ for $t = 1, \ldots, T$. Each task, $\mathcal{T}_t$, is comprised of a dataset $\mathcal{D}_t = \{(\boldsymbol{x}_i, y_i)\}_{i=1}^{N_t}$ which a model needs to learn to make predictions with. More generally, tasks are denoted by distinct tuples comprised of the conditional and marginal data distributions, $\{p_t(y|\mathbf{x}), p_t(\mathbf{x})\}$. After task $\mathcal{T}_t$ the model will lose access to the training dataset but its performance will be continually evaluated on all tasks $\mathcal{T}_i$ for $i \leq t$. We decompose predictors as $g = h \circ f$ such that $\hat{y} = g(\boldsymbol{x})$. We define $f$ as an embedding function mapping $f : \mathcal{X} \to \mathcal{Z}$ and $h$ as a head mapping to outputs $h : \mathcal{Z} \to \mathcal{Y}$. Some continual learning methods use a separate head per task $\{h_i\}_{i=1}^{T}$, these methods are called multi-headed while those that use one head are called single-headed.

### 2.2 Bayesian Continual Learning

We consider a setting in which task data arrives sequentially at time steps, $t = 1, 2, \ldots, T$. At the first time step, $t = 1$, the model parameterized by $\boldsymbol{\theta}$ receives the dataset $\mathcal{D}_1$ and learns the conditional distribution $p(y_i|\boldsymbol{x}_i, \boldsymbol{\theta})$ for all $(\boldsymbol{x}_i, y_i) \in \mathcal{D}_1$ ($i$ indexes a datapoint), the parameters $\boldsymbol{\theta}$ have a prior distribution $p(\boldsymbol{\theta})$. The posterior predictive distribution for a test point, $\boldsymbol{x}_1^*$ is:

$$p(y_1^*|\boldsymbol{x}_1^*, \mathcal{D}_1) = \int p(y_1^*|\boldsymbol{x}_1^*, \boldsymbol{\theta})p(\boldsymbol{\theta}|\mathcal{D}_1)d\boldsymbol{\theta}. \tag{1}$$

Computing this posterior predictive distribution above requires $p(\boldsymbol{\theta}|\mathcal{D}_1)$. For $t = 2$, a CL model is required to fit $p(y_i|\boldsymbol{x}_i, \boldsymbol{\theta})$ for $\mathcal{D}_1 \cup \mathcal{D}_2$. The posterior predictive distribution for a new test point $\boldsymbol{x}_2^*$ point is:

$$p(y_2^*|\boldsymbol{x}_2^*, \mathcal{D}_1, \mathcal{D}_2) = \int p(y_2^*|\boldsymbol{x}_2^*, \boldsymbol{\theta})p(\boldsymbol{\theta}|\mathcal{D}_1, \mathcal{D}_2)d\boldsymbol{\theta}. \tag{2}$$

The posterior must thus be updated to reflect this new conditional distribution. We can use repeated application of Bayes' rule to calculate the posterior distributions $p(\boldsymbol{\theta}|\mathcal{D}_1, \ldots, \mathcal{D}_T)$ as:

$$p(\boldsymbol{\theta}|\mathcal{D}_1, \ldots, \mathcal{D}_{T-1}, \mathcal{D}_T) = \frac{p(\mathcal{D}_T|\boldsymbol{\theta})p(\boldsymbol{\theta}|\mathcal{D}_1, \ldots, \mathcal{D}_{T-1})}{p(\mathcal{D}_T|\mathcal{D}_1, \ldots, \mathcal{D}_{T-1})}. \tag{3}$$

In the CL setting we lose access to previous training datasets: however, using repeated applications of Bayes' rule Eq. (3), allows us to sequentially incorporate information from past tasks in the parameters $\boldsymbol{\theta}$. At $t = 1$, we have access to $\mathcal{D}_1$ and the posterior over weights is:

$$\log p(\boldsymbol{\theta}|\mathcal{D}_1) = \log p(\mathcal{D}_1|\boldsymbol{\theta}) + \log p(\boldsymbol{\theta}) - \log p(\mathcal{D}_1). \tag{4}$$

At $t = 2$, we require a posterior $p(\boldsymbol{\theta}|\mathcal{D}_1, \mathcal{D}_2)$ to calculate the posterior predictive distribution in Eq. (2). However, we have lost access to $\mathcal{D}_1$. According to Bayes' rule, the posterior may be written as:

$$\log p(\boldsymbol{\theta}|\mathcal{D}_1, \mathcal{D}_2) = \log p(\mathcal{D}_2|\boldsymbol{\theta}) + \log p(\boldsymbol{\theta}|\mathcal{D}_1) - \log p(\mathcal{D}_2|\mathcal{D}_1), \tag{5}$$

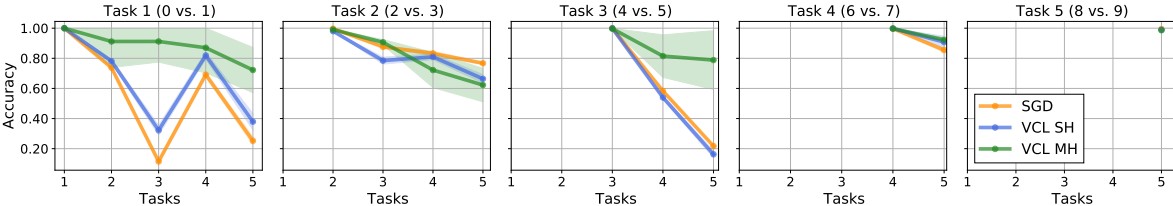

Figure 1: Accuracy on Split-MNIST for various CL methods with a two-layer BNN, all accuracies are an average and standard deviation over 10 runs with different random seeds. We compare a NN trained with SGD (single-headed) with VCL. We consider single-headed (SH) and multi-head (MH) VCL variants.

where we used the conditional independence of $\mathcal{D}_2$ and $\mathcal{D}_1$ given $\boldsymbol{\theta}$. We note that the likelihood is only dependent upon the current task dataset, $\mathcal{D}_2$, and that the prior encodes parameter knowledge from the previous task. Hence, we can use the posterior at $t$ as a prior for learning a new task at $t+1$. From Eq. (5) we require that our model with parameters $\boldsymbol{\theta}$ is a sufficient statistic of $\mathcal{D}_1$, making the likelihood conditionally independent of $\mathcal{D}_1$ given $\boldsymbol{\theta}$. This observation motivates the use of high-capacity predictors, such as Bayesian neural networks, that are flexible enough to learn $\mathcal{D}_1$.

### 2.2.1 Continual Learning Example: Split-MNIST

For the MNIST dataset (LeCun et al., 1998) we know that if we were to train a BNN we would achieve good performance by inferring the posterior $p(\boldsymbol{\theta}|\mathcal{D})$ and integrating out the posterior to infer the posterior predictive over a test point Eq. (1). So if we were to split the dataset MNIST into 5 two class classification tasks then we should be able to recursively recover the multi-task posterior $p(\boldsymbol{\theta}|\mathcal{D}) = p(\boldsymbol{\theta}|\mathcal{D}_1 \ldots, \mathcal{D}_5)$ using Eq. (5). This problem is called Split-MNIST (Zenke et al., 2017), where the first task involves the classification of the digits $\{0, 1\}$ then the second task classification of the digits $\{2, 3\}$ and so on.

We can define a 3 different CL settings Hsu et al. (2018); Van de Ven & Tolias (2019); van de Ven et al. (2022). When we allow the CL agent to make predictions with a task identifier $\tau$ the scenario is referred to as *task-incremental*. The identifier $\tau$ could be used to select different heads Section 2.1, for instance. This scenario is not compatible with sequential Bayesian inference outlined in Eq. (5) since no task identifier is required for making predictions. *Domain-incremental* learning is another scenario that doesn't have access to $\tau$ during evaluation and requires the CL agent to perform classification to the same output space for each task, for example for Split-MNIST the output space is $\{0, 1\}$ for all tasks, so this amounts to classifying between even and odd digits. Domain incremental learning is compatible with sequential Bayesian inference with a Bernoulli likelihood. The third scenario is *class-incremental* learning which also doesn't have access to $\tau$ but the agent needs to classify each example to its corresponding class. For Split-MNIST, for example, the output space is $\{0, \ldots, 9\}$ for each task. Class-incremental learning is compatible with sequential Bayesian inference with a categorical likelihood.

### 2.3 Variational Continual Learning

Variational CL (VCL; Nguyen et al. (2018)) simplifies the Bayesian inference problem in Eq. (5) into a sequence of approximate Bayesian updates on the distribution over random neural network weights $\boldsymbol{\theta}$. To do so, VCL uses the variational posterior from previous tasks as a prior for new tasks. In this way, learning to solve the first task entails finding a variational distribution $q_1(\boldsymbol{\theta}|\mathcal{D}_1)$ that maximizes a corresponding variational objective. For the subsequent task, the prior is chosen to be $q_1(\boldsymbol{\theta}|\mathcal{D}_1)$, and the goal becomes to learn a variational distribution $q_2(\boldsymbol{\theta}|\mathcal{D}_2)$ that maximizes a corresponding variational objective under this prior. Denoting the recursive posterior inferred from multiple datasets by $q_t(\boldsymbol{\theta}|\mathcal{D}_{1:t})$, we can express the variational CL objective for the $t$-th task as:

$$\mathcal{L}(\boldsymbol{\theta}, \mathcal{D}_t) = \mathbb{D}_{\mathrm{KL}}\left[q_t(\boldsymbol{\theta})||q_{t-1}(\boldsymbol{\theta}|\mathcal{D}_{1:t-1})\right] - \mathbb{E}_{q_t}[\log p(\mathcal{D}_t|\boldsymbol{\theta})]. \tag{6}$$

When applying VCL to the problem of Split-MNIST Figure 1, we can see that single-headed VCL barely performs better than SGD when remembering past tasks. Multi-headed VCL performs better, despite not being a requirement from sequential Bayesian inference Eq. (5). So why does single-head VCL not improve over SGD if we can recursively build up an approximate posterior using Eq. (5)? We hypothesize that it could be due to using a variational approximation of the posterior and so we are not actually strictly performing the Bayesian CL process described in Section 2.2. We test this hypothesis in the next section by propagating the true BNN posterior to verify whether we can recursively obtain the true multi-task posterior and so improve on single-head VCL and prevent catastrophic forgetting.

## 3  Bayesian Continual Learning with Hamiltonian Monte Carlo

To perform inference over BNN weights we use the HMC algorithm (Neal et al., 2011). We then use these samples and learn a density estimator that can be used as a prior for a new task[1]. HMC is considered the gold standard in approximate inference and is guaranteed to asymptotically produce samples from the true posterior[2]. we use posterior samples of $\boldsymbol{\theta}$ from HMC and then fit a density estimator over these samples, to use as a prior for a new task. This allows us to use a multi-modal posterior distribution over $\boldsymbol{\theta}$. In contrast, to a diagonal Gaussian variational posterior like in VCL. More concretely, to propagate the posterior $p(\boldsymbol{\theta}|\mathcal{D}_1)$ we use a density estimator, defined $\hat{p}(\boldsymbol{\theta}|\mathcal{D}_1)$, to fit a probability density on HMC samples as a posterior. For the next task $\mathcal{T}_2$ we can use $\hat{p}(\boldsymbol{\theta}|\mathcal{D}_1)$ as a prior for a new HMC sampling chain and so on (see Fig. 2). The density estimator priors need to satisfy two key conditions for use within HMC sampling. Firstly, that they are a probability density function. Secondly, that they are differentiable with respect to the input samples.

We use a toy dataset (Fig. 3) with two classes and inputs $\boldsymbol{x} \in \mathbb{R}^2$ (Pan et al., 2020). Each task is a binary classification problem where the decision boundary extends from left to right for each new task. We train a two layer BNN, with hidden state size of 10. We use a Gaussian Mixture Models (GMM) as a density estimator for approximating the posterior with HMC samples. We also tried Normalizing Flows which should be more flexible (Dinh et al., 2016) however these did not work robustly for HMC sampling[3]. To the best of our knowledge we are the first to incorporate flexible priors into the sampling methods like HMC.

Training a BNN with HMC on the same multi-task dataset gets a test accuracy of 1.0. Thus, the final posterior is suitable for continual learning under Eq. (3) we should be able to recursively arrive at the multi-task posterior with our recursive inference method with HMC. The results from Fig. 3

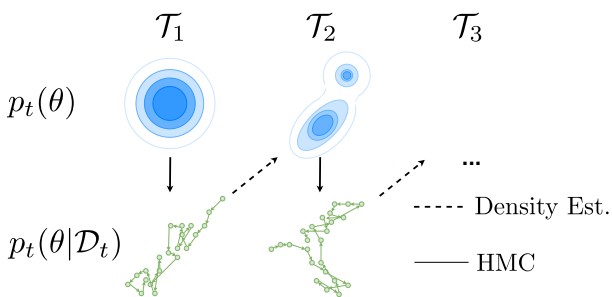

Figure 2: Illustration of the posterior propagation process; priors in blue are in the top row and posterior samples on the bottom row. This is a two step process where we first perform HMC with an isotropic Gaussian prior for $\mathcal{T}_1$ then perform density estimation on the HMC samples from the posterior to obtain $\hat{p}_1(\theta|\mathcal{D}_1)$. This posterior can then be used as a prior for the new task $\mathcal{T}_2$ and so on.

demonstrate that using HMC with an approximate multi-modal posterior fails to prevent forgetting and is less effective than using multi-head VCL. In fact, multi-head VCL clearly outperforms HMC indicating that the source of the knowledge retention is not through the propagation of the posterior but through the task-specific heads. For $\mathcal{T}_2$ we use $\hat{p}(\boldsymbol{\theta}|\mathcal{D}_1)$ instead of $p(\boldsymbol{\theta}|\mathcal{D}_1)$ as a prior and this will bias the HMC sampling for all subsequent tasks. In the next paragraph we detail the measures taken to ensure that our HMC chains have converged so we are sampling from the true posterior. Also we access the fidelity of the GMM density

---

[1]We considered Sequential Monte Carlo, but it is unable to scale to the dimensions required for the NNs we consider (Chopin et al., 2020). HMC on the other hand has recently been successfully scaled to relatively small BNNs of the size considered in this paper (Cobb & Jalaian, 2021) and ResNet models but at large computational cost (Izmailov et al., 2021).

[2]In the NeurIPS 2021 Bayesian Deep Learning Competition, the goal was to find an approximate inference method that is as "close" as possible to the posterior samples from HMC.

[3]RealNVP was very sensitive to the choice of random seed, the samples from the learned distribution did not give accurate predictions for the current task and led to numerical instabilities when used as a prior within HMC sampling.

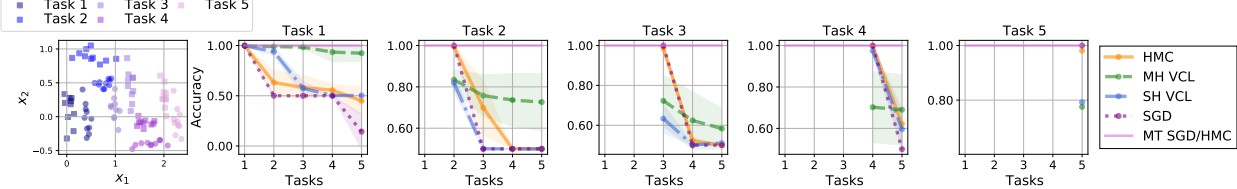

Figure 3: On the left is the toy dataset of 5 distinct 2-way classification tasks which involve classifying circles and squares (Pan et al., 2020). Also, continual learning binary classification test accuracies over 10 seeds. The pink solid line is a multi-task (MT) baseline accuracy using SGD/HMC with the same model as for the CL experiments.

estimator respect to the HMC samples. We also repeated these experiments with another toy dataset of five binary classification tasks where we observe similar results (Fig. 7).

For HMC we ensure that we are sampling from the posterior by assessing chain convergence and effective sample sizes (Fig. 11). The effective sample size measures the autocorrelation in the chain. The effective sample sizes for the HMC chains for our BNNs are similar to the literature (Cobb & Jalaian, 2021). Also, we ensure that the GMM approximate posterior is multi-modal and so has a more complex posterior in comparison to VCL, and that the GMM samples produce equivalent results to HMC samples for the current task (Fig. 10). See Appendix B for details.

The 2-d benchmarks we consider in this section are from previous works and are domain-incremental continual learning problems. The domain incremental setting is also simpler (van de Ven et al., 2022) than the class-incremental setting and thus a good starting point when attempting to perform exact sequential Bayesian inference. Despite this, we are not able to perform sequential Bayesian inference in BNNs despite using HMC which is considered the gold standard of Bayesian deep learning. HMC and density estimation with a GMM produces richer, accurate, and multi-modal posteriors. Despite this, we are still not able to sequentially build up the multi-task posterior or get much better results than an isotropic Gaussian posterior like single-head VCL. The weak point of this method is the density estimation, the GMM removes probability mass over areas of the BNN weight space posterior which is important for the new task. This demonstrates just how difficult a task it is to model BNN weight posteriors. In the next section, we study a different analytical example of sequential Bayesian inference and look at how model misspecification and task data imbalances can cause forgetting in Bayesian CL.

## 4 Bayesian Continual Learning and Model Misspecification

We now consider a simple analytical example where we can perform the sequential Bayesian inference Eq. (3) in closed form using conjugacy. We consider a simple setting where data points arrive online, one after another.

Observations $y_1, y_2, \ldots, y_t$ arrive online, each observation is generated by a hidden variable $\theta_1, \theta_2, \ldots, \theta_t \sim p$ where $p$ is a probability density function. At time $t$ we wish to infer the *filtering distribution* $p(\theta_t | y_1, y_2, \ldots, y_t)$ (Doucet et al., 2001) using sequential Bayesian inference, similarly to the *Kalman filter* (Kalman, 1960). The likelihood is $p(y_t | \theta_t) = \mathcal{N}(y_t; f(\,\cdot\,; \theta_t), \sigma^2)$ such that $y_t = f(\,\cdot\,; \theta_t) + \epsilon$ where $\epsilon \sim \mathcal{N}(0, \sigma^2)$ and $f(\,\cdot\,; \theta_t) = \theta_t$. We consider a Gaussian prior over the mean parameters $\theta$ such that

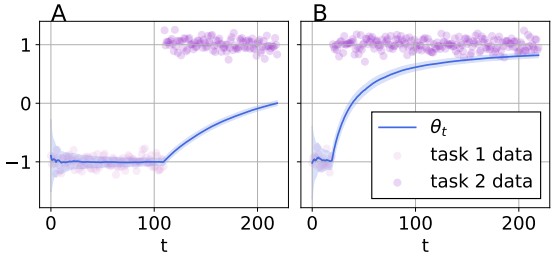

Figure 4: Posterior estimate of the filtering distribution Eq. (7) for two different scenarios with two tasks or changepoint.

$p(\theta_0) = \mathcal{N}(\theta_0; 0, \sigma_0^2)$. Since the conjugate prior for the mean is also Gaussian, the prior and posterior are $\mathcal{N}(\theta_{t-1}; \hat{\theta}_{t-1}, \hat{\sigma}_{t-1}^2)$ and $\mathcal{N}(\theta_t; \hat{\theta}_t, \hat{\sigma}_t^2)$. By using sequential Bayesian inference we can have closed-form update

equations for our posterior parameters:

$$\hat{\theta}_t = \hat{\sigma}_t^2 \left( \frac{y_t}{\sigma^2} + \frac{\hat{\theta}_{t-1}}{\hat{\sigma}_{t-1}^2} \right) = \hat{\sigma}_t^2 \left( \sum_{i=1}^{t} \frac{y_i}{\sigma^2} + \frac{\hat{\theta}_0}{\hat{\sigma}_0^2} \right), \qquad \frac{1}{\hat{\sigma}_t^2} = \frac{1}{\sigma^2} + \frac{1}{\hat{\sigma}_{t-1}^2}. \tag{7}$$

From Equation (7) the posterior mean follows a Gaussian distribution where the posterior mean is a sum of the online observation and the online prior. So the posterior mean is a weighted sum of the data. So, if the observations are non-stationary: if there is task change (more commonly referred to as a changepoint in the time-series literature). Then the mean parameter will encapsulate a global mean over both tasks rather than a mean for each task Fig. 4. Concretely, for task 1 the dataset is generated according to $\mathcal{N}(-1, \sigma^2)$ so we want the model to regress to this task. For task 2 the data is generated according to $\mathcal{N}(1, \sigma^2)$ and so we want our continual learning agent to regress well to this task too, afterwards. As with all continual learning benchmarks we require our model to perform equally well on both tasks at the end of training at $t = 220$. The model is clearly misspecified since a single parameter cannot regress to both of these tasks together. *Despite performing exact inference a misspecified model can forget*, Fig. 4. In the case of HMC we verified that our Bayesian neural network had perfect performance on all tasks *beforehand*. In Section 3 we had a well specified model but struggled with exact sequential Bayesian inference Eq. (3). With this 1-d online learning scenario we are performing exact inference, however we have a misspecified model. It is important to disentangle model misspecification and exact inference, and highlight that model misspecification is a caveat which has not been highlighted in the CL literature as far as we are aware. Furthermore we can only ensure that our models are well specified if we have access to data from all tasks a priori. So in the scenario of *online continual learning* (Aljundi et al., 2019b;a; De Lange et al., 2019) we cannot know if our model will perform well on all past and future tasks without making assumptions on the task distributions.

## 5 Sequential Bayesian Inference and Imbalanced Task Data

Neural Networks are complex models with a broad hypothesis space and hence are a suitably well-specified model when tackling continual learning problems (Wilson & Izmailov, 2020). However, we struggle to fit the posterior samples from HMC to perform sequential Bayesian inference in Section 3.

We continue to use Bayesian filtering and assume a Bayesian NN where the posterior is Gaussian with a full covariance. By modeling the entire covariance we enable modeling how each individual weight varies with respect to all others. We do this by interpreting online learning in Bayesian NNs as filtering (Ciftcioglu & Türkcan, 1995). Our treatment is similar to Aitchison (2020) who derives an optimizer by

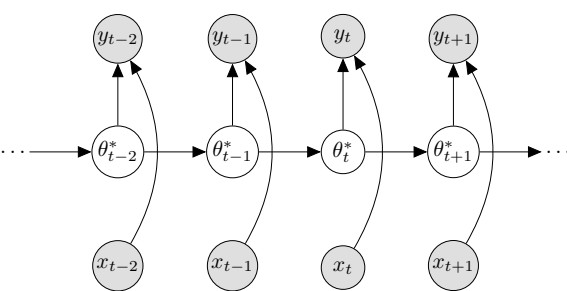

Figure 5: Graphical model for filtering. Grey and white nodes are observed and latent variables respectively.

leveraging Bayesian filtering. We consider inference in the graphical model depicted in Fig. 5. The aim is to infer the optimal BNN weights, $\boldsymbol{\theta}_t^*$ at $t$ given a single observation and the BNN weight prior. The previous BNN weights are used as a prior for inferring the posterior BNN parameters. We consider the online setting where a single data point $(\boldsymbol{x}_t, y_t)$ is observed at a time.

Instead of modeling the full covariance we instead consider each parameter $\theta_i$ as a function of all the other parameters $\boldsymbol{\theta}_{-it}$. We also assume that the values of the weights are close to those of the previous timestep (Jacot et al., 2018). To obtain the update equations for BNN parameters given a new observation and prior we make two simplifying assumptions as follows.

**Assumption 5.1.** *For a Bayesian neural network with output $f(\boldsymbol{x}_t; \boldsymbol{\theta})$ and likelihood $\mathcal{L}(\boldsymbol{x}_t, y_t; \boldsymbol{\theta})$, the derivative evaluated at $\boldsymbol{\theta}_t$ is $\mathbf{z}_t = \partial \mathcal{L}(\boldsymbol{x}_t, y_t; \boldsymbol{\theta}) / \partial \boldsymbol{\theta}|_{\boldsymbol{\theta}=\boldsymbol{\theta}_t}$ and the Hessian is $\boldsymbol{H}$. We assume a quadratic loss for a data point $(\boldsymbol{x}_t, y_t)$ of the form:*

$$\mathcal{L}(\boldsymbol{x}_t, y_t; \boldsymbol{\theta}) = \mathcal{L}_t(\boldsymbol{\theta}) = -\frac{1}{2} \boldsymbol{\theta}^\top \boldsymbol{H} \boldsymbol{\theta} + \mathbf{z}_t^\top \boldsymbol{\theta}, \tag{8}$$

*the result of a second-order Taylor expansion. The Hessian is assumed to be constant with respect to $(\boldsymbol{x}_t, y_t)$ (but not with respect to $\boldsymbol{\theta}$).*

To construct the dynamical equation for $\boldsymbol{\theta}$, consider the gradient for the $i$-th weight while all other parameters are set to their current estimate at the optimal value for the $\theta_{it}^*$:

$$\theta_{it}^* = -\frac{1}{H_{ii}}\boldsymbol{H}_{-ii}^\top\boldsymbol{\theta}_{-it}, \tag{9}$$

since $z_{it} = 0$ at a mode. The equation above shows us that the dynamics of the optimal weight $\theta_{it}^*$ is dependent on all the other current values of the parameters $\boldsymbol{\theta}_{-it}$. The dynamics of $\boldsymbol{\theta}_{-it}$ will be a complex stochastic process dependent on many different variables: such as the dataset, model architecture, learning rate schedule, etc.

**Assumption 5.2.** *Since reasoning about the dynamics of $\boldsymbol{\theta}_{-it}$ are intractable, we assume that at the next time-step the optimal weights are close to the previous timesteps with a discretized Ornstein-Uhlenbeck process for the weights $\boldsymbol{\theta}_{-it}$ with reversion speed $\vartheta \in \mathbb{R}_+$ and noise variance $\eta_{-i}^2$:*

$$p(\boldsymbol{\theta}_{-i,t+1}|\boldsymbol{\theta}_{-i,t}) = \mathcal{N}((1-\vartheta)\boldsymbol{\theta}_{-it}, \eta_{-i}^2), \tag{10}$$

*this implies that the dynamics for the optimal weight is defined by*

$$p(\theta_{i,t+1}^*|\theta_{i,t}^*) = \mathcal{N}((1-\vartheta)\theta_{it}^*, \eta^2), \tag{11}$$

*where $\eta^2 = \eta_{-i}^2\boldsymbol{H}_{-ii}^\top\boldsymbol{H}_{-ii}$.*

In simple terms, in Assumption 5.2 we assume a parsimonious model of the dynamics. That the next value of $\boldsymbol{\theta}_{-i,t}$ is close to their previous value according to a Gaussian, similarly to Aitchison (2020).

**Lemma 5.3.** *Under Assumptions 5.1 and 5.2 the dynamics and likelihood are Gaussian. Thus we are able to infer the posterior distribution over the optimal weights using Bayesian updates and by linearizing the BNN the update equations for the posterior of the mean and variance of the BNN for a new data point are:*

$$\mu_{t,post} = \sigma_{t,post}^2\left(\frac{\mu_{t,prior}}{\sigma_{t,prior}^2(\eta^2)} + \frac{y_t}{\sigma^2}g(\boldsymbol{x}_t)\right) \quad and \quad \frac{1}{\sigma_{t,post}^2} = \frac{g(\boldsymbol{x}_t)^2}{\sigma^2} + \frac{1}{\sigma_{t,prior}^2(\eta^2)}, \tag{12}$$

*where we drop the notation for the $i$-th parameter, the posterior is $\mathcal{N}(\theta_t^*; \mu_{t,post}, \sigma_{t,post}^2)$ and $g(\boldsymbol{x}_t) = \frac{\partial f(\boldsymbol{x}_t; \theta_{it}^*)}{\partial \theta_{it}^*}$ and $\sigma_{t,prior}^2$ is a function of $\eta^2$.*

See Appendix E for the derivation of Lemma 5.3. From Eq. (12) we can notice that the posterior mean depends linearly on the prior and a data dependent term and so will behave similarly to our previous example in Section 4. Under Assumption 5.1 and Assumption 5.2 then if there is a data imbalance between tasks in Eq. (12), then the data dependent term will dominate the prior term if there is more data for the current task.

In Section 3 we showed that it is very difficult with current machine learning tools to perform sequential Bayesian inference for simple CL problems with small Bayesian NNs. When we disentangle Bayesian inference and model misspecification we show showed that misspecified models can forget despite exact Bayesian inference. The only way to ensure that our model is well specified is to show that the multi-task posterior produces reasonable posterior predictive distributions $p(y|\boldsymbol{x}, \mathcal{D}) = \int p(y|\boldsymbol{x}, \mathcal{D}, \boldsymbol{\theta})p(\boldsymbol{\theta}|\mathcal{D})d\boldsymbol{\theta}$ for one's application. Additionally, in this section we have shown that if there is a task dataset size imbalance then we can get forgetting under certain assumptions.

# 6    Related Work

There has been a recent resurgence in the field of CL (Thrun & Mitchell, 1995) given the advent of deep learning. Methods which approximate sequential Bayesian inference Eq. (5) have been seminal in CL's

revival and have used a diagonal Laplace approximation (Kirkpatrick et al., 2017; Schwarz et al., 2018). The diagonal Laplace approximation have been enhanced by modelling covariances of between neural network weights in the same layer (Ritter et al., 2018). Instead of the Laplace approximation we can use a variational approximation for sequential Bayesian inference (Nguyen et al., 2018; Zeno et al., 2018). Using richer priors has also been explored (Ahn et al., 2019; Farquhar et al., 2020; Kessler et al., 2021; Mehta et al., 2021; Kumar et al., 2021; Loo et al., 2020). Gaussian processes have also been applied to CL problems leveraging inducing points to retain previous task functions (Titsias et al., 2020; Kapoor et al., 2021).

Bayesian methods which regularize weights have not matched up to the performance of experience replay-based CL methods (Buzzega et al., 2020) in terms of accuracy on CL image classification benchmarks. Instead of regularizing high-dimensional weight spaces, regularizing task functions is a more direct approach to combat forgetting (Benjamin et al., 2018). Bayesian NN weights can also be generated by a hypernetwork, where the hypernetwork needs only simple CL techniques to prevent forgetting (Henning et al., 2021). In particular, one can leverage the duality between the Laplace approximation and Gaussian Processes to develop a functional regularization approach to Bayesian CL (Swaroop et al., 2019) or using function-space variational inference (Rudner et al., 2022a;b).

In the next section, we propose a simple Bayesian continual learning baseline that models the data-generating continual learning process and performs exact sequential Bayesian inference in a low dimensional embedding space. Previous work has explored modeling the data-generating process by inferring the joint distribution of inputs and targets $p(\mathbf{x}, \mathbf{y})$ and learning a generative model to replay data to prevent forgetting (Lavda et al., 2018), and by learning a generative model per class and evaluating the likelihood of the inputs given each class $p(\mathbf{x}|\mathbf{y})$ (van de Ven et al., 2021).

# 7 Prototypical Bayesian Continual Learning

We have shown that sequential Bayes over NN parameters is very difficult (Section 3), and is only suitable for situations where the multi-task posterior is suitable for all tasks. We now show that a more fruitful approach is to model the full data-generating process of the CL problem and we propose a simple and scalable approach for doing so. In particular, we represent classes by prototypes (Snell et al., 2017; Rebuffi et al., 2017) to prevent catastrophic forgetting. We refer to this framework as Prototypical Bayesian Continual Learning, or ProtoCL for short. This approach can be viewed as a probabilistic variant of iCarl (Rebuffi et al., 2017), which creates embedding functions for different classes which are simply class means and predictions are made by nearest neighbors. ProtoCL also bears similarities to the few-shot learning model Probabilistic Clustering for Online Classification (Harrison et al., 2020), developed for few-shot image classification.

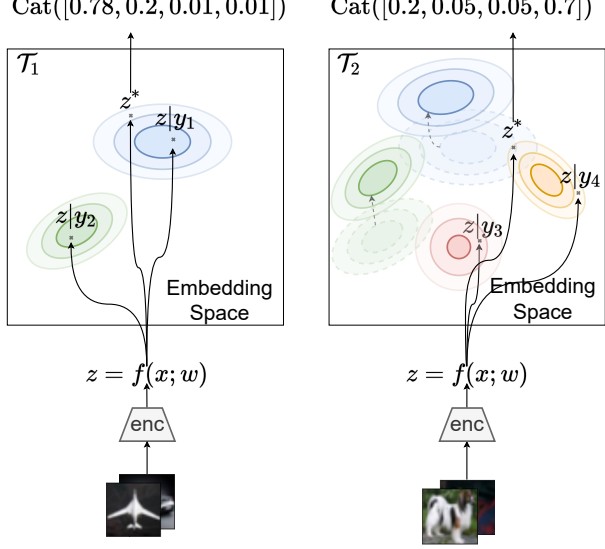

Figure 6: Overview of ProtoCL.

**Model.** ProtoCL models the generative CL process. We consider classes $j \in \{1, \ldots, J\}$, generated from a categorical distribution with a Dirichlet prior:

$$y_{i,t} \sim \text{Cat}(p_{1:J}), \quad p_{1:J} \sim \text{Dir}(\alpha_t). \tag{13}$$

Images are embedded into a embedding space by an encoder, $\boldsymbol{z} = f(\boldsymbol{x}; \boldsymbol{w})$ with parameters $\boldsymbol{w}$. The per class embeddings are Gaussian whose mean has a prior which is also Gaussian:

$$\boldsymbol{z}_{it}|y_{it} \sim \mathcal{N}(\bar{\boldsymbol{z}}_{yt}, \Sigma_\epsilon), \quad \bar{\boldsymbol{z}}_{yt} \sim \mathcal{N}(\boldsymbol{\mu}_{yt}, \Lambda_{yt}^{-1}). \tag{14}$$

See Fig. 6 for an overview of the model. To alleviate forgetting in CL, ProtoCL uses a coreset of past task data to continue to embed past classes distinctly as prototypes. The posterior distribution over class probabilities $\{p_j\}_{j=1}^J$ and class embeddings $\{\bar{z}_{y_j}\}_{j=1}^J$ is denoted in short hand as $p(\boldsymbol{\theta})$ with parameters $\eta_t = \{\alpha_t, \boldsymbol{\mu}_{1:J,t}, \Lambda_{1:J,t}^{-1}\}$. ProtoCL models each class prototype but does not use task specific NN parameters or modules like multi-head VCL. By modeling a probabilistic model over an embedding space this allows us to use powerful embedding functions $f(\,\cdot\,; \boldsymbol{w})$ without having to parameterize them probabilistically and so this approach will be more scalable than VCL, for instance.

**Inference.** As the Dirichlet prior is conjugate with the Categorical distribution and likewise the Gaussian over prototypes with a Gaussian prior over the prototype mean, we can calculate posteriors in closed form and update the parameters $\eta_t$ as new data is observed without using gradient based updates. We optimize the model by maximizing the posterior predictive distribution and use a softmax over class probabilities to perform predictions. We perform gradient-based learning of the NN embedding function $f(\,\cdot\,; \boldsymbol{w})$ and update the parameters, $\eta_t$ at each iteration of gradient descent as well, see Algorithm 1.

**Sequential updates.** We can obtain our parameter updates for the Dirichlet posterior by Categorical-Dirichlet conjugacy:

$$\alpha_{t+1,j} = \alpha_{t,j} + \sum_{i=1}^{N_t} \mathbb{I}(y_t^i = j), \tag{15}$$

where $N_t$ are the number of points seen during the update at time step $t$. Also, due to Gaussian-Gaussian conjugacy the posterior for the Gaussian prototypes is governed by:

$$\Lambda_{y_{t+1}} = \Lambda_{y_t} + N_y \Sigma_\epsilon^{-1} \tag{16}$$

$$\Lambda_{y_{t+1}} \boldsymbol{\mu}_{y_{t+1}} = N_y \Sigma_\epsilon^{-1} \bar{\boldsymbol{z}}_{y_t} + \Lambda_{y_t} \boldsymbol{\mu}_{y_t}, \, \forall y_t \in C_t, \tag{17}$$

where $N_y$ are the number of samples of class $y$ and $\bar{\boldsymbol{z}}_{y_t} = (1/N_y) \sum_{i=1}^{N_y} z_{yi}$, see Appendix D.2 for the detailed derivation.

**Objective.** We optimize the posterior predictive distribution of the prototypes and classes:

$$p(\boldsymbol{z}, y) = \int p(\boldsymbol{z}, y | \boldsymbol{\theta}_t; \eta_t) p(\boldsymbol{\theta}_t; \eta_t) d\boldsymbol{\theta}_t = p(y) \prod_{i=1}^{N_t} \mathcal{N}(\boldsymbol{z}_{it} | y_{it}; \boldsymbol{\mu}_{y_t,t}, \Sigma_\epsilon + \Lambda_{y_t,t}^{-1}). \tag{18}$$

Where the $p(y) = \alpha_y / \sum_{j=1}^J \alpha_j$, see Appendix D.3 for the detailed derivation. This objective can then be optimized using gradient based optimization for learning the prototype embedding function $\boldsymbol{z} = f(\boldsymbol{x}; \boldsymbol{w})$.

**Predictions.** To make a prediction for a test point $\boldsymbol{x}^*$ the class with the maximum (log)-posterior predictive is chosen, where the posterior predictive is:

$$p(y^* = j | \boldsymbol{x}^*, \boldsymbol{x}_{1:t}, y_{1:t}) = p(y^* = j | \boldsymbol{z}^*, \boldsymbol{\theta}_t) = \frac{p(y^* = j, \boldsymbol{z}^* | \boldsymbol{\theta}_t)}{\sum_i p(y = i, \boldsymbol{z}^* | \boldsymbol{\theta}_t)}, \tag{19}$$

see Appendix D.4 for further details.

**Preventing forgetting.** As we wish to retain the class prototypes. We make use of coresets: experience from previous tasks. At the end of learning a task $\mathcal{T}_t$, we retain a subset $\mathcal{M}_t \subset \mathcal{D}_t$ and augment each new task dataset to ensure that posterior parameters $\eta_t$ and prototypes are able to retain previous task information.

**Class-incremental learning.** In this CL setting we do not tell the CL agent which task it is being evaluated on with a task identifier $\tau$. So we cannot use the task identifier to select a specific head to use for classifying a test point, for example. Also, we require the CL agent to identify each class, $\{0, \ldots, 9\}$ for Split-MNIST and Split-CIFAR10 for example and not just $\{0, 1\}$ as in domain-incremental learning. Class-incremental learning is more general, realistic, and harder a problem setting and thus important to focus on rather than other settings, despite domain-incremental learning also being compatible with sequential Bayesian inference as described in Eq. (5).

---

**Algorithm 1** ProtoCL continual learning

---

1: **Input:** task datasets $\mathcal{T}_{1:T}$ , initialize embedding function: $f(\cdot; \boldsymbol{w})$, coreset: $\mathcal{M} = \emptyset$.
2: **for** $\mathcal{T}_1$ **to** $\mathcal{T}_T$ **do**
3:     **for** each batch in $\mathcal{T}_i \cup \mathcal{M}$ **do**
4:         Optimize $f(\cdot; \boldsymbol{w})$ by maximizing the posterior predictive $p(\mathbf{z}, y)$ Eq. (18)
5:         Obtain posterior over $\boldsymbol{\theta}$ by updating $\eta$, Eqs. (15) to (17).
6:     **end for**
7:     Add random subset from $\mathcal{T}_i$ to $\mathcal{M}$.
8: **end for**

---

**Implementation.** For Split-MNIST and Split-FMNIST the baselines and ProtoCL all use two layer NNs with a hidden state size of 200. For Split-CIFAR10 and Split-CIFAR100, the baselines and ProtoCL use a four layer convolution neural network with two fully connected layers of size 512 similarly to Pan et al. (2020). For ProtoCL and all baselines which rely on replay we fix the size of the coreset to 200 points per task. For all ProtoCL models we allow the prior Dirichlet parameters to be learned and set their initial value to 0.7 found by a random search over MNIST with ProtoCL. An important hyperparameter for ProtoCL is the embedding dimension of the Gaussian prototypes for Split-MNIST and Split-FMNIST this was set to 128 while for the larger vision datasets, this was set to 32 found using grid-search.

**Results.** ProtoCL produces good results on CL benchmarks on par or better than S-FSVI (Rudner et al., 2022b) which is state-of-the-art among Bayesian CL methods while being a lot more efficient to train and without requiring expensive variational inference. ProtoCL can flexibly scale to larger CL vision benchmarks producing better results than S-FSVI. Code to reproduce all experiments can be found here anonymous.4open.science/r/bayes_cl_exploration. All our experiments are in the more realistic class incremental learning setting, which is a harder setting than those reported in most CL papers, so the results in Table 1 are lower for certain baselines than in the respective papers. We use 200 data points per task, see Figure 12 for a sensitivity analysis of the performance over the Split-MNIST benchmark as a function of core size for ProtoCL.

The stated aim of ProtoCL is not to provide a novel state-of-the-art method for CL, but rather to propose a simple baseline that takes an alternative route than weight-space sequential Bayesian inference. We can achieve strong results that mitigate forgetting, namely by modeling the generative CL process and using sequential Bayesian inference over a few parameters in the class prototype embedding space. We argue that modeling the generative CL process is a fruitful direction for further research rather than attempting sequential Bayesian inference over the weights of a BNN. ProtoCL scales to 10 tasks of Split-CIFAR100 which to the best of our knowledge, is the most number of tasks and classes which has been considered by previous Bayesian continual learning methods.

Table 1: Mean accuracies across all tasks over CL vision benchmarks for *class incremental learning* (Van de Ven & Tolias, 2019). All results are averages and standard errors over 10 seeds. *Uses the predictive entropy to make a decision about which head for class incremental learning.

| Method | Coreset | Split-MNIST | Split-FMNIST |
|---|---|---|---|
| VCL (Nguyen et al., 2018) | ✗ | $33.01 \pm 0.08$ | $32.77 \pm 1.25$ |
|    + coreset | ✓ | $52.98 \pm 18.56$ | $61.12 \pm 16.96$ |
| HIBNN* (Kessler et al., 2021) | ✗ | $85.50 \pm 3.20$ | $43.70 \pm 20.21$ |
| FROMP (Pan et al., 2020) | ✓ | $84.40 \pm 0.00$ | $68.54 \pm 0.00$ |
| S-FSVI (Rudner et al., 2022b) | ✓ | $\mathbf{92.94 \pm 0.17}$ | $80.55 \pm 0.41$ |
| ProtoCL (**ours**) | ✓ | $\mathbf{93.73 \pm 1.05}$ | $\mathbf{82.73 \pm 1.70}$ |

Table 2: Mean accuracies across all tasks over CL vision benchmarks for *class incremental learning* (Van de Ven & Tolias, 2019). All results are averages and standard errors over 10 seeds. *Uses the predictive entropy to make a decision about which head for class incremental learning. Training times have been benchmarked using an Nvidia RTX3090 GPU.

| Method | Training time (sec) ($\downarrow$) | Split CIFAR-10 (acc) ($\uparrow$) |
|---|---|---|
| FROMP (Pan et al., 2020) | $1425 \pm 28$ | $48.92 \pm 10.86$ |
| S-FSVI (Rudner et al., 2022b) | $44434 \pm 91$ | $50.85 \pm 3.87$ |
| ProtoCL (**ours**) | $\mathbf{384 \pm 6}$ | $\mathbf{55.81 \pm 2.10}$ |
| | | Split CIFAR-100 (acc) |
| S-FSVI (Rudner et al., 2022b) | $37355 \pm 1135$ | $20.04 \pm 2.37$ |
| ProtoCL (**ours**) | $\mathbf{1425 \pm 28}$ | $\mathbf{23.96 \pm 1.34}$ |

## 8 Discussion & Conclusion

In this paper, we have revisited the use of sequential Bayesian inference for CL. We can use sequential Bayes to recursively build up the multi-task posterior Eq. (5). Previous methods have relied on approximate inference and see little benefit over SGD. We test the hypothesis of whether this poor performance is due to the approximate inference scheme by using HMC in two simple CL problems. HMC asymptotically samples from the true posterior and we use a density estimator over HMC samples to use as a prior for a new task within the HMC sampling process. This density is multi-modal and accurate with respect to the current task but is not able to improve over using an approximate posterior. This demonstrates just how challenging it is to work with BNN weight posteriors. The source of error comes from the density estimation step. We then look at an analytical example of sequential Bayesian inference where we perform exact inference however due to model misspecification, we observe forgetting. The only way to ensure a well specified model is to assess the multi-task performance over all tasks a priori. This might not be possible in online CL settings. We then model an analytical example over Bayesian NNs and under certain assumptions show that if there is task data imbalances then this will cause forgetting. Because of these results, we argue against performing weight space sequential Bayesian inference and instead model the generative CL problem. We introduce a simple baseline called ProtoCL. ProtoCL doesn't require complex variational optimization and achieves competitive results to state-of-the-art in the realistic setting of class incremental learning.

This conclusion should not be a surprise since the latest Bayesian CL papers have all relied multi-head architectures or inducing points/coresets to prevent forgetting, rather than better weight-space inference schemes. Our observations are in line with recent theory from (Knoblauch et al., 2020) which states that optimal CL requires perfect memory. Although the results were shown with deterministic NNs the same results follow for BNN with a single set of parameters. Future research directions include enabling coresets of task data to efficiently and accurately approximate the posterior of a BNN to remember previous tasks.

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
