# OpenReview forum: "On Sequential Bayesian Inference for Continual Learning"
_TMLR — Rejected by TMLR_

### Review · Reviewer_Vh1k · 2023-01-10

**Summary Of Contributions:**

The authors consider the task of continual learning via Bayesian inference. Within their work, they first discuss a series of problems that arise in this problem setting centred around the problem of catastrophic forgetting and demonstrate that this still arises even if we have access to (approximatively) the true posterior via HMC samples or can evaluate the posterior analytically, but use a misspecified model. The authors propose an approach they term _Prototypical Bayesian Continual Learning (ProtoCL)_. It relies on deterministically embedding observed features within a representation space, within which they rely on a mixture of Gaussians, allowing for analytical posterior updates due to a conjugacy structure. Catastrophic forgetting is avoided in this setting, by retaining random prototypes of prior tasks and thus their associated Gaussian posteriors.

**Audience:**

Yes

**Broader Impact Concerns:**

None.

**Claims And Evidence:**

Yes

**Requested Changes:**

Improvement in/Discussion of the remarks given above in the weaknesses section.

**Strengths And Weaknesses:**

# Strengths and Weaknesses
## Strengths
- The authors propose a simple approach to improve upon prior work in the Bayesian continual learning literature and demonstrate its efficiency both in terms of accuracy as well as runtime over a sequence of several experiments.
- The series of observations raised in this paper make for a good empirical evaluation to raise awareness of problems that can arise within Bayesian approaches to the task of CL.

## Weaknesses
- Figure 1 and its relatives will probably be difficult to understand to a reader not familiar with the literature as they lack a proper explanation of the specific setup. Including a short one in the main text/appendix will be helpful.
- Split-MNIST and its cousins are mentioned but not properly introduced and cited as having been proposed in prior work already
- The bibliography contains a lot of references to arXiv preprints that are instead published papers.

### Minor
- Q1: Is the SGD standard error in Figure 1 just too small or is it lacking an error bar?
- Q2: Can the authors explain why VCL with both the SH and MH struggles so much with solving the rather simple tasks in Fig 3?
- Q3: The size of the random subset which is retained (Alg 1, line 7) is not mentioned for the different experiments. Can the authors comment on their size as well as how sensitive the method is to this hyperparameter?
- Q4: Mentions that the initial $\alpha$ parameters are learned via gradient descent. This translates into essentially a Type II ML/Empirical Bayes approach. What is the performance difference between learning them this way and keeping them fixed at the value suggested by the random search approach?
- (Very) Minor comment on footnote 1: While HMC has indeed been scaled to BNNs, Cobb & Jalaian (2021) still rely on relatively small networks, while Izmailov et al. (2021) require enormous computational cost to be able to do so. In its current variant, the footnote reads as if HMC for BNNs were a solved task.
- Fig 13 seems to lack the error bars claimed to be present in the caption.



## Typos
- p2, denoted T for t=1,...**,** T has a missing comma
- Fig 3 appears before Fig 2
- g changes its definition from Sec 2.1 to Lemma 5.3
- p9: We retain **a** subset $M_t$
- p9: first Split-CIFAR10 misses the `-'
- p10: is not **to** provide a novel

---

> ### Author Response · Authors · 2023-01-17
> **Author response to Reviewer Vh1k (1/2)**
>
> Thank you for reviewing our manuscript and for your constructive feedback.
>
> > Figure 1 and its relatives will probably be difficult to understand to a reader not familiar with the literature as they lack a proper explanation of the specific setup. Including a short one in the main text/appendix will be helpful.
>
> > Split-MNIST and its cousins are mentioned but not properly introduced and cited as having been proposed in prior work already
>
> We have introduced Split-MNIST at the end of Section 2.2 where we introduce Bayesian Continual Learning (CL) and give Split-MNIST as an example. Thanks to your comment we have now added a formal introduction of the Split-MNIST experiment to make this clearer for the reader by adding a new subsection (2.2.1) in the updated manuscript. We will also add the appropriate citation for the introduction of the Split-MNIST experiment (i.e., [1]).
>
> [1] Zenke, Friedemann, Ben Poole, and Surya Ganguli. "Continual learning through synaptic intelligence." International Conference on Machine Learning. PMLR, 2017.
>
> > The bibliography contains a lot of references to arXiv preprints that are instead published papers.
>
> Thanks for pointing this out. We have gone through all the references again and updated them to the associated conference proceedings or journals where possible.
>
> > Q2: Can the authors explain why VCL with both the SH and MH struggles so much with solving the rather simple tasks in Fig 3?
>
> The baseline MH VCL network retains the performance on the first task as well as ensuring that this decision boundary is retained across all tasks, however performance on the other tasks is inconsistent. This is due to the fact that the variational distribution is regularized against a highly certain (i.e., low variance) empirical prior distribution when learning a new task (Eq. 6). This a strong regularization constraint and prevents the model from achieving good performance on new tasks. MH VCL’s heads ensure that performance on the first task is retained since the heads are not updated when learning new tasks. However, they do act as a prior for the new task head. For the SH VCL baseline, performance is poor, since it doesn’t use separate heads for each task and so this explicit form of knowledge retention through separate parameters doesn’t allow it to retain knowledge of the first task decision boundary. On top of that the variational distribution is being regularized against a highly certain prior (i.e., low variance) and so learning a new task decision boundary is also not optimal for the new task. This observation, of inconsistent and poor performance from VCL, was also seen in the FROMP paper on the same dataset, Figure 6 [2], where the authors use an additional coreset of points to retain previous task information. We do not study the use of the coreset since it is not a requirement when using sequential Bayesian inference. Our VCL experiments use the implementation from the original authors [3].
>
> [2] Pan, Pingbo, et al. "Continual deep learning by functional regularisation of memorable past." Advances in Neural Information Processing Systems 33 (2020): 4453-4464.
>
> [3] Nguyen, Cuong V., et al. "Variational Continual Learning." International Conference on Learning Representations. 2018.
>
> > Q3: The size of the random subset which is retained (Alg 1, line 7) is not mentioned for the different experiments. Can the authors comment on their size as well as how sensitive the method is to this hyperparameter?
>
> For all our experiments we use 200 points per task. This is mentioned in our Implementation paragraph in Section 7. So the memory will be of size 1000 for the CL experiments with 5 tasks and 2000 for the CL experiments which have 10 tasks. We included a sensitivity analysis in the appendix (Figure 12), where we vary the size of the memory buffer over all tasks and measure ProtoCL’s accuracy. As expected, ProtoCL is sensitive to the size of the memory buffer, since the memory ensures that past class prototypes are remembered.

---

> > ### Author Response · Authors · 2023-01-17
> > **Author response to Reviewer Vh1k (2/2)**
> >
> > > Q4: Mentions that the initial $\alpha$ parameters are learned via gradient descent. This translates into essentially a Type II ML/Empirical Bayes approach. What is the performance difference between learning them this way and keeping them fixed at the value suggested by the random search approach?
> >
> > We did not find any significant difference between learning the Dirichlet prior parameters $\alpha$ using Empirical Bayes versus keeping the value fixed at 0.78. When we fix the $\alpha$ to 0.78 we obtain accuracies of $92.71 \pm 0.90$ Split-MNIST over 5 seeds which is not significantly different to the values which we report in Table 1 of $93.73 \pm 1.05$. Also when we fix the $\alpha$ to 0.1 and 1.0 we obtain accuracies of $93.67 \pm 0.89$ and $93.67 \pm 0.94$ respectively (with 5 different random seeds) which also doesn’t show a significant difference in the accuracies. This is the expected behavior as the Bayesian updates will very quickly update the Dirichlet parameters $\alpha$ when learning from the data, Eq 15 of the manuscript, and the prior will be less important the more data we see.
> >
> > > (Very) Minor comment on footnote 1: While HMC has indeed been scaled to BNNs, Cobb & Jalaian (2021) still rely on relatively small networks, while Izmailov et al. (2021) require enormous computational cost to be able to do so. In its current variant, the footnote reads as if HMC for BNNs were a solved task.
> >
> > You are correct, using HMC for BNNs is computationally expensive and is difficult to currently scale to large Bayesian neural networks. This needs to be emphasized here. We have rephrased the footnote to mention these two caveats for each reference.
> >
> > > Fig 13 seems to lack the error bars claimed to be present in the caption.
> >
> > Thanks for the question, there are indeed error bars but they are very small since there is little variation between different runs with different random seeds.
> >
> > Many thanks for pointing out typos in the manuscript - this is very helpful and we have updated the manuscript accordingly.

---

### Review · Reviewer_epht · 2023-01-16

**Summary Of Contributions:**

This paper starts by studying sequential Bayesian inference on the parameters of a neural network as an approach for continual learning and preventing catastrophic forgetting. First, using two simple datasets (Split MNIST and a 2-way classification problem in 2D-space taken from Pan et al., 2020 NeurIPS), it is shown that variational continual learning (VCL) does not perform well when performed with a single-headed output layer. The authors then show for the toy problem in 2D-space that even when they use better, gold-standard approximations to the posterior distributions (i.e., Hamiltonian Monte Carlo followed by density estimation using GMMs), they are unable to substantially improve upon the performance of VCL. The authors take this as evidence that sequential Bayesian inference for continual learning is very challenging. Next, the authors raise another problem with sequential Bayesian inference. Using a simple filtering toy problem in which they can perform exact Bayesian inference, the authors show that if the model is misspecified (e.g., a wrong choice of prior), the model might still not perform well. The authors also claim that in the case of data unbalance, a misspecified model can even forget despite performing exact inference.

In the final part of the paper, motivated by these failure cases of sequential Bayesian inference, the authors propose that a more fruitful approach for continual learning is to model the full data-generating process. To this end the authors propose Prototypical Bayesian Continual Learning (ProtoCL), which learns a generative classifier by modeling each class as a Gaussian in an embedding space of an underlying neural network. To prevent drift in the underlying neural network, data from previous tasks are stored and replayed. The proposed ProtoCL method outperforms a number of Bayesian continual learning methods on several class-incremental learning benchmarks (Split-MNIST, Split-FMNIST, Split CIFAR-10, Split CIFAR-100).

**Audience:**

Yes

**Broader Impact Concerns:**

No concerns in this regard.

**Claims And Evidence:**

No

**Requested Changes:**

- Clarify that sequential exact Bayesian inference with a correctly specified model should be able to solve any continual learning problem, and rephrase several potentially misleadingly formulated hypotheses (e.g., “we test whether having access to the true posterior is guaranteed to prevent catastrophic forgetting” in the abstract).
- Remove the claim that a misspecified model that performs exact Bayesian inference can forget (or convince me that this claim is true in a meaningful way). I would probably even suggest the authors to leave out most of sections 4 and 5.
- Clarify and explain/justify the different types of continual learning that are studied throughout the paper.
- Remove or properly justify claims about “state-of-the-art”.
- Discuss existing work in the continual learning literature that has also proposed and demonstrated that modeling the data-generating process is a fruitful approach for continual learning (for example the paper on the Generative Classifier; van de Ven et al., 2021 CVPR-W)
- [optional] Compare against iCaRL and/or discuss why it is important to have a probabilistic version of iCaRL


**Strengths And Weaknesses:**

**Strengths**

I think this is an interesting, and potentially important and impactful paper. Sequential Bayesian inference on the parameters of a deep neural network is an important, often-used approach in continual learning that underlies many existing methods. Demonstrating fundamental problems with this approach is therefore an important contribution. I think the authors succeed in making a convincing case that sequential Bayesian inference on the parameters of a neural network is very challenging in practice.

I also consider the proposed ProtoCL method a useful contribution to the continual learning literature. In particular, I think the point that modeling the generative continual learning process is a fruitful direction for further research is insightful and convincing, although it should be said that a similar point has been made in the continual learning literature before (see below).


**Weaknesses**

As the paper is currently presented, it is not always clear that the problems with sequential Bayesian inference highlighted by the authors are, ultimately, practical in nature. Unless I’m mistaken, sequential exact Bayesian inference starting from a suitable prior should still be able to solve any continual learning problem. The contribution of this paper is to provide evidence that there are severe practical problems with this approach, even when you try really hard on a relatively simple toy problem.
One source of potential confusion stems from several potentially misleadingly formulated questions / hypotheses. For example, in the abstract the authors state that they “test whether having access to the true posterior is guaranteed to prevent catastrophic forgetting”. Perhaps strictly speaking this formulation is not incorrect, but it is meaningless because having access to the true posterior by itself of course does not provide any guarantees (so this question could be answered without any experimentation). To use the true posterior the authors do need to make approximations, and I think it is critical that the authors are clear and upfront about this (among others in the abstract) to avoid misinterpretation.

I think the claim that a misspecified model can forget despite performing exact inference is problematic (Sections 4 and 5). It is not explained very clearly what the authors mean by forgetting in this case, but it seems that the authors refer to Figure 4B, in which the posterior estimate of the theta parameter gradually moves towards task 2 upon seeing more and more data from task 2. But why is this “forgetting”? If an algorithm observes 90% data coming from task 2 and 10% data from task 1, the desired/expected behavior should be to bias one's predictions towards task 2, shouldn’t it? Let’s turn it around. What would happen if the sequence would be reversed: the first 90% of data presented belongs to task 2, the last 10% belongs to task 1. Unless I’m mistaken, the final state of the model would be exactly the same. But in this case it is clear that it can’t be claimed that task 1 has been forgotten.

A confusing aspect of this paper is that it studies three different types of incremental learning, without clearly discussing or acknowledging that it does. Sections 2 and 3 predominantly deal with domain-incremental learning problems, while section 7 deals with class-incremental learning problems. These are fundamentally different types of problems with different challenges (see Van de Ven et al, 2022 Nature Machine Intelligence). In fact, it seems to me that the ProtoCL approach proposed in section 7 is only really suited for class-incremental learning. If this approach would be applied to the problems studied in sections 2 and 3, I expect it would yield little benefit. Is there a reason that the authors do not apply their ProtoCL method to the problems studied in sections 2 and 3?
Another issue arising from mixing these different types of incremental learning relates to the comparisons in section 2 and 3. Here, VCL with a separate output layer per task (multi-headed or MH), which is task-incremental, is compared with doing sequential Bayesian inference on a network with a single output layer that is shared between tasks (single-headed or SH), which is domain-incremental. This is problematic because these are two different kinds of problems; for example, joint training on all tasks of the MH network will give a different result than joint training on all tasks of the SH network. So even sequential exact Bayesian inference on a SH network might not fully close the gap to a MH baseline. There is also a risk that the current presentation suggests that VCL with MH *does* work well. But I don’t think that is the message the authors want to convey?

Then some comments about the “Prototypical Bayesian Continual Learning” method that is proposed and evaluated in section 7.
Firstly, a claim is made that the method S-FSVI is “state-of-the-art on the smaller CL benchmarks”. If I look at Van de Ven et al. (2022, Nature Machine Intelligence), which also contains a comparison for class-incremental learning on Split MNIST, the approach “Generative Classifier” seems to perform better than S-FSVI (and also better or at least comparable to ProtoCL) without relying on coresets. I also expect that a method such as Dark Experience Replay (Buzzega, 2020 NeurIPS) will likely perform better then S-FSVI and potentially also better than ProtoCL. I appreciate that the authors state that the aim of ProtoCL is not to provide a novel state-of-the-art method, and that is fine, but then the authors should refrain from (implicitly) making claims about state-of-the-art.
Secondly, I think it is important that the authors acknowledge and discuss existing work in the continual learning literature that proposed and demonstrated that modeling the data-generating process is a fruitful approach for continual learning. For example, the above mentioned Generative Classifier rather explicitly takes this approach. In fact, the authors motivate their ProtoCL method by stating that modeling the full data-generating process of the CL problem is a fruitful approach; yet, the approach they then propose does not actually do this (they only model the generating process in the embedding space of a neural network) while the Generative Classifier does model the full data-generating process.

Finally, the results reported in Figure 12 seem to be at odds with the results of ProtoCL reported in Table 1. Could the authors comment on this?

---

> ### Author Response · Authors · 2023-01-27
> **Author response to Reviewer epht (1/n)**
>
> We would like to thank the reviewer for their thoughtful and constructive feedback. We respond inline.
>
> > Unless I’m mistaken, sequential exact Bayesian inference starting from a suitable prior should still be able to solve any continual learning problem.
>
> Assuming that by the “prior” we are talking about the model and the prior probability distribution over parameters. Then, we agree that it would be possible to sequentially build the posterior to solve any continual learning problem using, Eq 5. In practice we need to have access to the union of all task datasets to see if the multi-task posterior is suitable for solving all tasks first, to assess whether the multi-task posterior $p(\theta | D_{1}, …, D_{T})$ is suitable. For instance, we know that for the 2-d datasets in Fig 3 and Fig 7, and for Split-MNIST that if we use a 2-layer BNN on the union of all task datasets then we can obtain good performance. This means we have a suitable model and prior. In this case sequential exact Bayesian inference over the 2 layer BNN will solve these Split continual learning problems by sequentially building up the posterior.
>
> As we noted in the paper if our continual learning problem is online and we do not have access to the whole dataset apriori. We have no guarantee that we have a suitable model and prior and so even if we perform exact sequential Bayesian inference this might not prevent forgetting, Section 4.
>
> > For example, in the abstract the authors state that they “test whether having access to the true posterior is guaranteed to prevent catastrophic forgetting”. Perhaps strictly speaking this formulation is not incorrect, but it is meaningless because having access to the true posterior by itself of course does not provide any guarantees (so this question could be answered without any experimentation). To use the true posterior the authors do need to make approximations, and I think it is critical that the authors are clear and upfront about this (among others in the abstract) to avoid misinterpretation.
>
> Many thanks for drawing this to our attention. We agree and were careful with our language regarding “access” to the true posterior, which HMC is able to sample from. Which as you noted is not incorrect. We believe that access to the true posterior is meaningful as previous variational approaches, such as VCL, do not allow for this. We believe that by stating in the abstract that we are fitting a density estimator over HMC samples we are clear as to the approximation when doing sequential Bayesian inference, in Section 3.
>
> > I think the claim that a misspecified model can forget despite performing exact inference is problematic (Sections 4 and 5). It is not explained very clearly what the authors mean by forgetting in this case, but it seems that the authors refer to Figure 4B, in which the posterior estimate of the theta parameter gradually moves towards task 2 upon seeing more and more data from task 2. But why is this “forgetting”? If an algorithm observes 90% data coming from task 2 and 10% data from task 1, the desired/expected behavior should be to bias one's predictions towards task 2, shouldn’t it? Let’s turn it around. What would happen if the sequence would be reversed: the first 90% of data presented belongs to task 2, the last 10% belongs to task 1. Unless I’m mistaken, the final state of the model would be exactly the same. But in this case it is clear that it can’t be claimed that task 1 has been forgotten.
>
> In this simple 1-d example we haven’t elaborated on what the tasks are and how we define forgetting. We have now clarified this in the manuscript. We define each task as learning a model which regresses to each dataset. In task 1 the data is generated according to $N(-1, 0.1^2)$ so we want a model which regresses well to this task. For task 2 the data is generated according to $N(1, 0.1^2)$ and so we want our continual learning agent to regress well to this task too, afterwards. As in the case of classification we require the continual learning agent to perform predictions **equally well** on both tasks. If we consider continual learning metrics such as forgetting, forward and backward transfer [1, 2] they are all defined and averaged per task. So we need to perform equally well on all tasks despite there being a task dataset imbalance. We usually want to correct the task data imbalance in continual learning which could prevent learning a new task or increase forgetting [3].
>
> [1] Chaudhry, Arslan, et al. "Riemannian walk for incremental learning: Understanding forgetting and intransigence." Proceedings of the European Conference on Computer Vision (ECCV). 2018.
>
> [2] Lopez-Paz, David, and Marc'Aurelio Ranzato. "Gradient episodic memory for continual learning." Advances in neural information processing systems 30 (2017).
>
> [3] Chrysakis, Aristotelis, and Marie-Francine Moens. "Online continual learning from imbalanced data." International Conference on Machine Learning. PMLR, 2020.

---

> > ### Author Response · Authors · 2023-01-27
> > **Author response to Reviewer epht (2/n)**
> >
> > > A confusing aspect of this paper is that it studies three different types of incremental learning, without clearly discussing or acknowledging that it does. Sections 2 and 3 predominantly deal with domain-incremental learning problems, while section 7 deals with class-incremental learning problems. These are fundamentally different types of problems with different challenges (see Van de Ven et al, 2022 Nature Machine Intelligence). In fact, it seems to me that the ProtoCL approach proposed in section 7 is only really suited for class-incremental learning.
> >
> > The principal goal of the paper is to perform sequential Bayesian inference. Sequential Bayesian inference is compatible with both domain and class incremental learning with Bernoulli and Categorical likelihoods respectively, for classification problems. The simple 2-d benchmarks in Fig 3 [4] and Fig 7 [5] are originally domain incremental. Also, it is well known that continual learning models achieve better results on domain incremental learning than on class incremental. So practically this is also a good place to start for our HMC method. Domain incremental learning has a smaller output space (binary classification) while class-incremental learning has a larger output space and so is thus harder (multi-class classification).
> >
> > That ProtoCL is designed for class-incremental learning is exactly the point of the model. We advocate moving away from weight space sequential Bayesian inference and designing a solution that models the data-generating process of the class incremental learning problem directly. Using the data generating process to model domain incremental learning will result in a different model to ProtoCL as we will have multiple classes of images mapping to {0, 1} and so the model will be different. It is also worth saying that class-incremental learning is a more general, more realistic, and harder continual learning problem that as far as we are aware no other Bayesian continual learning methods have looked at, so this is a fruitful direction of work.
> >
> > > This is problematic because these are two different kinds of problems; for example, joint training on all tasks of the MH network will give a different result than joint training on all tasks of the SH network. So even sequential exact Bayesian inference on a SH network might not fully close the gap to a MH baseline
> >
> > We have done joint training on all tasks for the 2d datasets in Section 3 using HMC and the BNNs achieve good performance by inferring these posteriors. So in theory we should be able to use sequential Bayesian inference on each task to build up the posterior. No use of multiple heads is required. These are the pink lines in Fig 3 and Fig 7.
> >
> > > There is also a risk that the current presentation suggests that VCL with MH does work well. But I don’t think that is the message the authors want to convey?
> >
> > This is the correct conclusion from Fig 1. It seems that using different heads for each task is very effective for continual learning despite not being a requirement for sequential Bayesian inference. Our hypothesis is that the variational approximation to the posterior is too simple. So that’s why we use HMC with density estimation over samples to see whether we can obtain a posterior that better represents the true posterior and is able to yield better results.
> >
> > [4] Pan, Pingbo, et al. "Continual deep learning by functional regularisation of memorable past." Advances in Neural Information Processing Systems 33 (2020): 4453-4464.
> >
> > [5] Henning, Christian, et al. "Posterior meta-replay for continual learning." Advances in Neural Information Processing Systems 34 (2021): 14135-14149.

---

> > > ### Author Response · Authors · 2023-01-27
> > > **Author response to Reviewer epht (3/n)**
> > >
> > > > Firstly, a claim is made that the method S-FSVI is “state-of-the-art on the smaller CL benchmarks”. If I look at Van de Ven et al. (2022, Nature Machine Intelligence), which also contains a comparison for class-incremental learning on Split MNIST, the approach “Generative Classifier” seems to perform better than S-FSVI (and also better or at least comparable to ProtoCL) without relying on coresets. I also expect that a method such as Dark Experience Replay (Buzzega, 2020 NeurIPS) will likely perform better then S-FSVI and potentially also better than ProtoCL. I appreciate that the authors state that the aim of ProtoCL is not to provide a novel state-of-the-art method, and that is fine, but then the authors should refrain from (implicitly) making claims about state-of-the-art.
> > >
> > > We would like to highlight that ProtoCL is state-of-the-art among Bayesian continual learning methods. Our intention was not to state that our method is state-of-the-art among all continual learning models. We admit that it is confusing to state that ProtoCL is state-of-the-art without specifying that the comparison is restricted to other Bayesian methods. As there are other approaches, which you rightly comment perform better, we have removed statements that claim ProtoCL is state-of-the-art and make clear our comparisons are with Bayesian CL methods. We have made this statement clear in the abstract and added a sentence of explanation at the end of Section 7.
> > >
> > > > Secondly, I think it is important that the authors acknowledge and discuss existing work in the continual learning literature that proposed and demonstrated that modeling the data-generating process is a fruitful approach for continual learning.
> > >
> > > We have added a discussion and citations regarding previous work modeling the data-generating process for CL [6, 7] in the related works section of the manuscript.
> > >
> > > [6] van de Ven, Gido M., Zhe Li, and Andreas S. Tolias. "Class-incremental learning with generative classifiers." Proceedings of the IEEE/CVF Conference on Computer Vision and Pattern Recognition. 2021.
> > >
> > > [7] Lavda, Frantzeska, et al. "Continual classification learning using generative models." arXiv preprint arXiv:1810.10612 (2018).
> > >
> > > > In fact, the authors motivate their ProtoCL method by stating that modeling the full data-generating process of the CL problem is a fruitful approach; yet, the approach they then propose does not actually do this (they only model the generating process in the embedding space of a neural network) while the Generative Classifier does model the full data-generating process.
> > >
> > > Yes this is correct, ProtoCL uses a neural network encoder to extract features and define class prototypes in this feature space. By doing this, we are able to leverage the benefits of automatic feature extraction of neural networks in addition to closed-form Bayesian updates over latent variables in this lower dimensional feature space. It is also worth noting the Generative Classifier also takes advantage of pre-trained feature extractors and models a class conditional distribution $p(x|y)$ over features, so $x$ are the features of an encoder, as well as ProtoCL for similar reasons, see Section 4.3 in [8].
> > >
> > > [8] van de Ven, Gido M., Zhe Li, and Andreas S. Tolias. "Class-incremental learning with generative classifiers." Proceedings of the IEEE/CVF Conference on Computer Vision and Pattern Recognition. 2021.
> > >
> > >
> > > > Finally, the results reported in Figure 12 seem to be at odds with the results of ProtoCL reported in Table 1. Could the authors comment on this?
> > >
> > > Thanks for pointing this out. We have updated Fig 12 in the manuscript with optimal hyperparameters so that it is consistent with Table 1.
> > >
> > > > # Requested Changes:
> > > > Clarify that sequential exact Bayesian inference with a correctly specified model should be able to solve any continual learning problem, and rephrase several potentially misleadingly formulated hypotheses (e.g., “we test whether having access to the true posterior is guaranteed to prevent catastrophic forgetting” in the abstract).
> > >
> > > See our earlier response to your initial comments inline, above.

---

> > > > ### Author Response · Authors · 2023-01-27
> > > > **Author response to Reviewer epht (4/n, n = 4)**
> > > >
> > > > > Remove the claim that a misspecified model that performs exact Bayesian inference can forget (or convince me that this claim is true in a meaningful way). I would probably even suggest the authors to leave out most of sections 4 and 5.
> > > >
> > > > In Section 4, we show that despite performing exact Bayesian inference over a model we still observe forgetting. The model is simple and misspecified as we are using a single parameter regression model to regress to multiple tasks. A potential solution to this problem setting would be the introduction of an additional latent variable (such as a HMM) to condition the likelihood. However, the way the model is specified in Section 4 causes it to forget previous tasks. We highlight this setting since it is parallel to how the community uses BNNs in continual learning benchmark problems.
> > > >
> > > > In this simple 1-d example we define each task as learning a model which regresses to each dataset. In task 1 the data is generated according to $N(-1, 0.1^2)$ so we want a model which regresses well to this task. For task 2 the data is generated according to $N(1, 0.1^2)$ and so we want our continual learning agent to regress well to this task too, afterward. We require the continual learning agent to perform predictions equally well on both tasks at the end of training, at time $t=220$ in Fig 4. In Fig 4A we observe that after learning on both task datasets, at the end of training, at $t=220$, the model regresses poorly to both task 1 and task 2. So it has forgotten how to regress to task 1 and its prior has strongly regularized it and prevented it from learning task 2, as well (negative forward transfer). In Fig 4B, we show a different scenario where there is a task data imbalance and at the end of training, at $t=220$, the model has forgotten how to regress to the first task, but regresses well to task 2. This model is misspecified since a single parameter cannot regress to two different values, despite exact sequential Bayesian inference.
> > > >
> > > > > Clarify and explain/justify the different types of continual learning that are studied throughout the paper.
> > > >
> > > > Many thanks for bringing this to our attention. We have provided more clarity over the reasoning for the different types of continual learning studied in the manuscript in our previous answer to your comment in the weaknesses section of the review. We have also added a discussion elaborating on the different types of continual learning which we focus on in the manuscript in Section 2.2.1. We have justified the use of domain incremental learning as a suitable starting step for attempting to perform exact Bayesian sequential inference in Section 3 and justified our focus on class-incremental learning in Section 7, both setups are compatible with sequential Bayesian inference.
> > > >
> > > > > Remove or properly justify claims about “state-of-the-art”.
> > > >
> > > > We have responded to this comment above in our response to the weaknesses.
> > > >
> > > > > Discuss existing work in the continual learning literature that has also proposed and demonstrated that modeling the data-generating process is a fruitful approach for continual learning (for example the paper on the Generative Classifier; van de Ven et al., 2021 CVPR-W)
> > > >
> > > > Many thanks, we have added a discussion on prior works [6, 7] which model the continual learning data generating distribution.
> > > >
> > > > > [optional] Compare against iCaRL and/or discuss why it is important to have a probabilistic version of iCaRL.
> > > >
> > > > ProtoCL and iCaRL are similar insofar as iCaRL classifies points by using a nearest neighbor rule with respect to class mean embeddings. ProtoCL classifies a point according to the probability it was generated by a Gaussian class embedding. iCaRL also introduces a distillation loss and coreset construction heuristics to help it maintain previous task / class prototypes. ProtoCL maintains a simple coreset with equal number of points from each task.
> > > >
> > > > As to why it is important to have a probabilistic version, this is an excellent question and this goes to the heart of why Bayesian approaches are desirable. One perspective is that they are able to marginalize over many different model instances by integrating over the posterior rather than by using a single set of weights, this (should) provide more robust predictions in terms of accuracies and calibrations [9]. Another advantage is that we can robustly update our beliefs in the presence of new data. For instance, if you look at Fig 13 where the prior probability distribution over the class probabilities is updated in the presence of new data. This adds a certain amount of interpretability. This principle is at the heart of Bayesian continual learning as well.
> > > >
> > > > [9] Wilson, Andrew G., and Pavel Izmailov. "Bayesian deep learning and a probabilistic perspective of generalization." Advances in neural information processing systems 33 (2020): 4697-4708.

---

> > > > > ### Comment · Reviewer_epht · 2023-02-03
> > > > > **Reviewer response to author rebuttal**
> > > > >
> > > > > Thanks to the authors for responding to my review. Please find below some further comments.
> > > > >
> > > > > **“Many thanks for drawing this to our attention. We agree and were careful with our language regarding “access” to the true posterior, which HMC is able to sample from. Which as you noted is not incorrect. We believe that access to the true posterior is meaningful as previous variational approaches, such as VCL, do not allow for this.”**
> > > > >
> > > > > This does not address my concern that the stated “[we] test whether having access to the true posterior is guaranteed to prevent catastrophic forgetting” is meaningless. Simply having access to something does not provide any guarantees, as one can simply choose not to use it (or to use it in a naive way). This hypothesis is thus badly formulated and needs to be revised.
> > > > >
> > > > > Moreover, I now realize it is also not fully correct to say that by using Hamiltonian Monte Carlo access is provided to the true posterior. Perhaps this could be argued for the second task (although only in the limit of infinite samples, which should be clarified), but for any later task it is not true anymore, at least not with respect to the initial prior, due to the approximations made by fitting the density estimator. It seems to me this should be discussed.
> > > > >
> > > > >
> > > > > **“We believe that by stating in the abstract that we are fitting a density estimator over HMC samples we are clear as to the approximation when doing sequential Bayesian inference, in Section 3.”**
> > > > >
> > > > > This was not clear to me when I read the abstract, and I expect it would not be clear to others.
> > > > >
> > > > >
> > > > > **The example in section 4 does not show “forgetting”**
> > > > >
> > > > > The bias that is observed in this experiment towards task 2 is due to the imbalance in data, not due to the temporal order in which the data is presented. Furthermore, this bias towards task 2 is not due to misspecification of the prior; even if the prior would be specified correctly, imbalance data would lead to a bias towards the task with more data. Finally, as raised by another reviewer, in the context of deep neural networks, the issue of prior misspecification seems not very relevant.
> > > > >
> > > > >
> > > > > **“We would like to highlight that ProtoCL is state-of-the-art among Bayesian continual learning methods. Our intention was not to state that our method is state-of-the-art among all continual learning models. We admit that it is confusing to state that ProtoCL is state-of-the-art without specifying that the comparison is restricted to other Bayesian methods.”**
> > > > >
> > > > > On p10 of the revised manuscript there is still a claim about state-of-the-art without specification that this only considers Bayesian continual learning methods: “ProtoCL produces good results on CL benchmarks on par or better than S-FSVI (Rudner et al., 2022b) which is state-of-the-art on the smaller CL benchmarks”.
> > > > >
> > > > > If the authors want to make this claim with the specification that it only holds for “Bayesian continual learning methods”, the authors should include a definition of when a method can be considered a Bayesian continual learning method.
> > > > >
> > > > > This is particularly relevant because it seems to me that, in contrast to methods such as VCL, the Bayesian aspect of the proposed ProtoCL method is not actually relevant from a continual learning perspective. It seems to me the continual learning performance of this method comes from the use of replay for learning the feature extractor combined with learning a generative classifier in the resulting embedding space. That the parameters of this generative classifier are learned using sequential Bayesian inference might have some benefits, but as far as I can see not from a continual learning perspective.
> > > > >
> > > > >
> > > > > **Previous work has explored modeling the data-generating process by inferring the joint distribution of inputs and targets p(x, y) and learning a generative model to replay data to prevent forgetting using variational auto-encoders (Lavda et al., 2018; van de Ven et al., 2021).** (Added to p8 of the revised manuscript.)
> > > > >
> > > > > The approach described by Van de Ven et al. (2021) does not replay data.

---

> > > > > > ### Author Response · Authors · 2023-02-06
> > > > > > **Author response to reviewer (1/2)**
> > > > > >
> > > > > > Many thanks for the comments.
> > > > > >
> > > > > > > This does not address my concern that the stated “[we] test whether having access to the true posterior is guaranteed to prevent catastrophic forgetting” is meaningless. Simply having access to something does not provide any guarantees, as one can simply choose not to use it (or to use it in a naive way). This hypothesis is thus badly formulated and needs to be revised.
> > > > > >
> > > > > > We have updated our abstract so that instead of formulating this statement in the abstract as a hypothesis with strong language like “true posterior” and “guaranteed” we will state that we wish to revisit sequential Bayesian inference. We have removed the term guarantee as we understand your concern. We have changed the hypothesis “We revisit sequential Bayesian inference and test whether having access to the true posterior is guaranteed to prevent catastrophic forgetting in Bayesian neural networks” to “We revisit sequential Bayesian inference and assess whether using the an old task’s posterior as a prior for a new task can prevent catastrophic forgetting in Bayesian neural networks”.
> > > > > >
> > > > > > > Moreover, I now realize it is also not fully correct to say that by using Hamiltonian Monte Carlo access is provided to the true posterior. Perhaps this could be argued for the second task (although only in the limit of infinite samples, which should be clarified), but for any later task it is not true anymore, at least not with respect to the initial prior, due to the approximations made by fitting the density estimator. It seems to me this should be discussed.
> > > > > >
> > > > > > Thank you for mentioning this. We agree, when we attempt to propagate the true posterior that this is correct for the first task. For subsequent tasks, task 2 onwards, the prior is the approximate density estimate over HMC samples and so we cannot guarantee that HMC will sample from the true posterior since the prior is a fitted density estimate and there will be some error. We have added a sentence in Section 3 discussing this.
> > > > > >
> > > > > > >This was not clear to me when I read the abstract, and I expect it would not be clear to others.
> > > > > >
> > > > > > Thanks for bringing this to our attention have now rephrased the abstract to make this clearer. The sentence “We propagate the posterior as a prior for new tasks by fitting a density estimator on Hamiltonian Monte Carlo samples.” has been edited to “We propagate the posterior as a prior for new tasks by approximating the posterior via fitting a density estimator on Hamiltonian Monte Carlo samples”.
> > > > > >
> > > > > > >The bias that is observed in this experiment towards task 2 is due to the imbalance in data, not due to the temporal order in which the data is presented. Furthermore, this bias towards task 2 is not due to misspecification of the prior; even if the prior would be specified correctly, imbalance data would lead to a bias towards the task with more data.
> > > > > >
> > > > > > We agree that the bias due to task 2 is due to the imbalance and not the order, Eq 7. Also, we agree that the bias towards the second task is not due to the misspecification of the prior over the linear model parameters, $p(\theta)$. In the linear model in Section 4 there is no specification of the prior, $p(\theta)$ that will not lead to forgetting of the first task or no learning of the second task (if the prior is very restrictive).
> > > > > >
> > > > > > >Finally, as raised by another reviewer, in the context of deep neural networks, the issue of prior misspecification seems not very relevant.
> > > > > >
> > > > > > Our focus is on model misspecification in Section 4, not prior misspecification. We have not mentioned prior misspecification in our paper. The issue of prior misspecification e.g. using a Laplace prior versus a Gaussian prior over weights, is an interesting topic and left for future work.
> > > > > >
> > > > > > > On p10 of the revised manuscript there is still a claim about state-of-the-art without specification that this only considers Bayesian continual learning methods: “ProtoCL produces good results on CL benchmarks on par or better than S-FSVI (Rudner et al., 2022b) which is state-of-the-art on the smaller CL benchmarks”.
> > > > > >
> > > > > > Thanks for drawing this to our attention we have amended this in the manuscript.

---

> > > > > > > ### Author Response · Authors · 2023-02-06
> > > > > > > **Author response to reviewer (2/2)**
> > > > > > >
> > > > > > > >If the authors want to make this claim with the specification that it only holds for “Bayesian continual learning methods”, the authors should include a definition of when a method can be considered a Bayesian continual learning method.
> > > > > > >
> > > > > > > >This is particularly relevant because it seems to me that, in contrast to methods such as VCL, the Bayesian aspect of the proposed ProtoCL method is not actually relevant from a continual learning perspective. It seems to me the continual learning performance of this method comes from the use of replay for learning the feature extractor combined with learning a generative classifier in the resulting embedding space. That the parameters of this generative classifier are learned using sequential Bayesian inference might have some benefits, but as far as I can see not from a continual learning perspective.
> > > > > > >
> > > > > > > It is correct that the Bayesian aspect of ProtoCL is not the principle mechanism for enabling previous tasks to be remembered. We are clear that we must use a coreset of previous task data to ensure that previous class Gaussian embeddings are remembered. Our model is Bayesian insofar as the parameters in the embedding space are all learned via sequential Bayesian inference.
> > > > > > >
> > > > > > > VCL also uses a coreset of data (and task heads). With only sequential variational inference, VCL achieves poor performance, Fig 1, this is also seen in the literature [1]. VCL is widely regarded as a Bayesian continual learning method too. In Section 3 our HMC method only relies on sequential Bayesian inference and it still performs poorly despite our use of HMC and an accurate and multimodal density estimator as a posterior. This shows just how difficult it is to rely solely on Bayesian inference in NN weight space.
> > > > > > >
> > > > > > > [1] Farquhar, Sebastian, and Yarin Gal. "Towards robust evaluations of continual learning." arXiv preprint arXiv:1805.09733 (2018).
> > > > > > >
> > > > > > > > The approach described by Van de Ven et al. (2021) does not replay data.
> > > > > > >
> > > > > > > Thanks for drawing this to our attention we have updated the related works section.

---

> > > > > > > > ### Comment · Reviewer_epht · 2023-02-08
> > > > > > > > **Remaining concerns**
> > > > > > > >
> > > > > > > > Thank you for your response.
> > > > > > > >
> > > > > > > >
> > > > > > > > **Our focus is on model misspecification in Section 4, not prior misspecification. We have not mentioned prior misspecification in our paper.**
> > > > > > > >
> > > > > > > > Apologies for using the wrong term. Please replace in my previous comment “prior misspecification” by “model misspecification”. For my concern, it does not really matter whether the misspecification is in the prior or in the model.
> > > > > > > >
> > > > > > > >
> > > > > > > > **We agree that the bias due to task 2 is due to the imbalance and not the order.**
> > > > > > > >
> > > > > > > > Yet, in section 4 the term “forgetting” is still used. Is this intentional or has the pdf not been fully updated yet? If the order of the data is not relevant for the observed phenomenon, it surely can’t be called forgetting.
> > > > > > > >
> > > > > > > > Furthermore, it seems to me that the bias towards task 2 in Figure 4B is also not a result of model misspecification. Even if the model would be specified correctly, unless I'm mistaken, imbalance would still lead to a bias towards the task with more data. Yet the authors interpret this experiment as highlighting that model misspecification is the issue.
> > > > > > > > To further reinforce my concern, I think also the ProtoCL method would “suffer” from this kind of bias in the case of data imbalance (i.e., ProtoCL will be more likely to predict classes from which it has seen more training data).
> > > > > > > >
> > > > > > > > Finally, as also raised by another reviewer, the issue of model misspecification does not seem very relevant in the context of deep neural networks. Could the authors address this?
> > > > > > > >
> > > > > > > >
> > > > > > > > **It is correct that the Bayesian aspect of ProtoCL is not the principle mechanism for enabling previous tasks to be remembered. … Our model is Bayesian insofar as the parameters in the embedding space are all learned via sequential Bayesian inference.**
> > > > > > > >
> > > > > > > > Does the Bayesian aspect of ProtoCL provide a mechanism for remembering previous tasks at all? It seems to me that because the parameters in the embedding space are all class-specific, forgetting is prevented by design and learning these via sequential Bayesian inference does not contribute to this. If this is indeed the case, does it still make sense to call ProtoCL a Bayesian continual learning method? Or to only compare against Bayesian CL methods?

---

> > > > > > > > > ### Author Response · Authors · 2023-02-10
> > > > > > > > > **Response to remaining concerns (1/2)**
> > > > > > > > >
> > > > > > > > > Many thanks for your questions.
> > > > > > > > >
> > > > > > > > > > Apologies for using the wrong term. Please replace in my previous comment “prior misspecification” by “model misspecification”. For my concern, it does not really matter whether the misspecification is in the prior or in the model.
> > > > > > > > >
> > > > > > > > > > Finally, as also raised by another reviewer, the issue of model misspecification does not seem very relevant in the context of deep neural networks. Could the authors address this?
> > > > > > > > >
> > > > > > > > > Apologies, but it is not clear what your first point of concern alludes to. We will assume it refers to your second concern quoted (above). It also does matter whether your question is about model misspecification or prior misspecification because in Section 4 we are concerned with model misspecification and not prior misspecification.
> > > > > > > > >
> > > > > > > > > The issue of model misspecification in deep learning has seen a lot of interesting research recently [2, 3]. Popular model classes are for instance MLPs, CNNs and linear models. The class of MLPs is a very broad and flexible model class which supports a wide range of datasets. This is in contrast to the class of linear models which we explore in Section 4, see Fig 2 from [1]. So the issue of model misspecification is definitely less of an issue in MLPs, since they can model many datasets. Within the class of MLPs and CNNs different models or architectures themselves admit very different results [2]. However you could still think of a CL scenario where we have tabular data and image data in different tasks and a CNN or MLP by themselves will not be a good model for this CL scenario. Using a deep neural network is not a sufficient condition for preventing model misspecification. This is why we opt for modelling the data generating distribution of our continual learning problem under investigation. This is why we develop ProtoCL for class-incremental learning.
> > > > > > > > >
> > > > > > > > > [1] Wilson, Andrew G., and Pavel Izmailov. "Bayesian deep learning and a probabilistic perspective of generalization." Advances in neural information processing systems 33 (2020): 4697-4708.
> > > > > > > > >
> > > > > > > > > [2] Immer, Alexander, et al. "Scalable marginal likelihood estimation for model selection in deep learning." International Conference on Machine Learning. PMLR, 2021.
> > > > > > > > >
> > > > > > > > > [3] Lotfi, Sanae, et al. "Bayesian model selection, the marginal likelihood, and generalization." International Conference on Machine Learning. PMLR, 2022.
> > > > > > > > >
> > > > > > > > > > Yet, in section 4 the term “forgetting” is still used. Is this intentional or has the pdf not been fully updated yet? If the order of the data is not relevant for the observed phenomenon, it surely can’t be called forgetting.
> > > > > > > > >
> > > > > > > > > Apologies but the question is not clear here, we will do our best to clarify, however. The order in which the data is seen doesn’t affect the posterior distribution after seeing **all** the data sequentially. This can be seen in Eq 7 for the linear model as we are summing over all our data: the order doesn’t matter. This is the motivation for using sequential Bayesian inference over NNs since an ordering where data is chunked into tasks or randomly sampled will recover the same posterior.
> > > > > > > > >
> > > > > > > > > We will repeat our previous explanation to show why there is forgetting in our example in Section 4. In this simple 1-d example we define each task as learning a model which regresses to each dataset. In task 1 the data is generated according to $N(-1, 0.1^2)$ so we want a model which regresses well to this task. At $t=110$ (so after the first task), after seeing all the data from task 1 we can see that the linear model performs well on this task. For task 2 the data is generated according to $N(1, 0.1^2)$ and so we want our continual learning agent to regress well to this task too, afterwards. We require the continual learning agent to perform predictions equally well on both tasks at the end of training, at time $t=220$ in Fig 4. In Fig 4A we observe that after learning on both task datasets, that at the end of training, at $t=220$, the model regresses poorly to both task 1 and task 2. It regresses to around 0 for both tasks. **So it has forgotten how to regress to task 1** and its prior has strongly regularized it and prevented it from learning task 2, as well.

---

> > > > > > > > > > ### Author Response · Authors · 2023-02-10
> > > > > > > > > > **Response to remaining concerns (2/2)**
> > > > > > > > > >
> > > > > > > > > > > Furthermore, it seems to me that the bias towards task 2 in Figure 4B is also not a result of model misspecification. Even if the model would be specified correctly, unless I'm mistaken, imbalance would still lead to a bias towards the task with more data. Yet the authors interpret this experiment as highlighting that model misspecification is the issue.
> > > > > > > > > > To further reinforce my concern, I think also the ProtoCL method would “suffer” from this kind of bias in the case of data imbalance (i.e., ProtoCL will be more likely to predict classes from which it has seen more training data).
> > > > > > > > > >
> > > > > > > > > > The model which we use is a linear model, which by definition can only model a single task, in Section 4. We wish our model to regress to multiple tasks, so the model is not fit for this continual learning task; it is misspecified. On the other hand, if we had a linear model which could regress to multiple tasks then we would have a well-specified model. Such a set of models could be a latent variable linear model where $t$ is the task index, and the distribution over parameters is $p(\theta_t | z_t) = N(\mu_t, \sigma_t^2)$, where the latent variable $Z_t$ is from a discrete distribution over tasks $p(z_t) = Cat(p_t)$. The likelihood would then be $p(y | \theta, z)$ and this model will be able to regress to multiple values depending on which $z$ is inferred. So this model is well-specified for the problem, even with the data imbalance. To recap, Bayesian inference, Eq 3, will perform inference over the parameters of any model even if the class of models isn’t appropriate for a dataset or continual learning problem. If the model is not well specified then we will get poor performance (which can entail forgetting), Section 4.
> > > > > > > > > >
> > > > > > > > > > ProtoCL (and many other CL methods) will need a method to correct for the dataset imbalance such as using reservoir sampling [1]. The well-specified model we define above won’t need this since it infers the task it needs to solve. The purpose of our discussion regarding dataset imbalances is to show that relying solely on sequential Bayesian inference over parameters of a (misspecified) linear model. Or under certain assumptions those of a Bayesian NN, then this imbalance will bias inference of these parameters towards a certain task.
> > > > > > > > > >
> > > > > > > > > > [1] Vitter, Jeffrey S. "Random sampling with a reservoir." ACM Transactions on Mathematical Software (TOMS) 11.1 (1985): 37-57.
> > > > > > > > > >
> > > > > > > > > > > Does the Bayesian aspect of ProtoCL provide a mechanism for remembering previous tasks at all? It seems to me that because the parameters in the embedding space are all class-specific, forgetting is prevented by design and learning these via sequential Bayesian inference does not contribute to this. If this is indeed the case, does it still make sense to call ProtoCL a Bayesian continual learning method? Or to only compare against Bayesian CL methods?
> > > > > > > > > >
> > > > > > > > > >
> > > > > > > > > > ProtoCL requires a coreset of past tasks to maintain previous class embeddings otherwise the NN encoder will forget how to embed past tasks. Without the coreset, the encoder will embed previous task data differently without the Gaussian class embeddings being updated in tandem with the encoder. We have a class-specific Gaussian but the mean embedding encoder is shared by all tasks, Eq 17. This will cause ProtoCL to be unable to classify previous classes correctly without the memory. Just because we use a coreset to maintain previous class embeddings to enable sequential Bayesian updates in an embedding space does not make our method any less Bayesian. Consider FROMP [1] and S-FSVI [2] both of these approaches perform functional sequential Bayesian updates which require a coreset so that previous task functions can be regularized. Both of these approaches are considered Bayesian or probabilistic and both require a coreset like ProtoCL. So just because we require a coreset doesn’t make our method non-Bayesian.
> > > > > > > > > >
> > > > > > > > > > We show that by modeling the data-generating process from a Bayesian perspective with ProtoCL, we can improve over sequential Bayesian inference over NN weights. Therefore, we want to compare to methods that perform sequential Bayesian inference over weights. We also want to compare against methods that perform sequential Bayesian inference in function space as these are state-of-the-art amongst Bayesian methods.
> > > > > > > > > >
> > > > > > > > > > [1] Pan, Pingbo, et al. "Continual deep learning by functional regularisation of memorable past." Advances in Neural Information Processing Systems 33 (2020): 4453-4464.
> > > > > > > > > >
> > > > > > > > > > [2] Rudner, Tim GJ, et al. "Continual Learning via Sequential Function-Space Variational Inference." International Conference on Machine Learning. PMLR, 2022.

---

> > > > > > > > > > > ### Comment · Reviewer_epht · 2023-02-15
> > > > > > > > > > > **Remaining concerns not yet addressed**
> > > > > > > > > > >
> > > > > > > > > > > Thank you for the response. I’m afraid your answers do not address the core of my remaining concerns. Please find below some further comments that will hopefully help to make my concerns more clear.
> > > > > > > > > > >
> > > > > > > > > > >
> > > > > > > > > > >
> > > > > > > > > > > **The order in which the data is seen doesn’t affect the posterior distribution after seeing** *all* **the data sequentially.**
> > > > > > > > > > >
> > > > > > > > > > > OK, let’s focus on the point in time after which all data has been seen in the example of Figure 4B. (But in principle the same can be done for any point in time.) If I understand your claim correctly, you claim that at this point in time forgetting has taken place of task 1. But at the same time you agree that performance on both tasks would be exactly the same if the order of the data had been reversed (i.e., exactly the same data up to that point is seen but in a different order: now first task 2 and then task 1). To me this indicates that the lower performance on task 1 at this point in time cannot, in a meaningful way, be attributed to or described as forgetting.
> > > > > > > > > > >
> > > > > > > > > > >
> > > > > > > > > > >
> > > > > > > > > > > **ProtoCL (and many other CL methods) will need a method to correct for the dataset imbalance such as using reservoir sampling [1].**
> > > > > > > > > > >
> > > > > > > > > > > Does this mean that the issue that you identify in section 4 and 5 is not addressed by the method you propose in section 7?
> > > > > > > > > > >
> > > > > > > > > > >
> > > > > > > > > > >
> > > > > > > > > > > **The well-specified model we define above won’t need this since it infers the task it needs to solve.**
> > > > > > > > > > >
> > > > > > > > > > > I do not think this is the case, unless you assume that task inference is flawless, which is an important assumption.
> > > > > > > > > > >
> > > > > > > > > > >
> > > > > > > > > > >
> > > > > > > > > > > **So just because we require a coreset doesn’t make our method non-Bayesian.**
> > > > > > > > > > >
> > > > > > > > > > > This was not my concern. My concern is that the Bayesian aspect of the method you propose (i.e. the sequential Bayesian inference of the parameters in the embedding space) does not contribute to the incremental learning *at all*.

---

> > > > > > > > > > > > ### Comment · Reviewer_epht · 2023-02-15
> > > > > > > > > > > > **Official recommendation needs to be submitted**
> > > > > > > > > > > >
> > > > > > > > > > > > I’m afraid that I now need to submit my official recommendation for this paper. As indicated in my original review, I think this paper makes a valuable contribution in the form of making a convincing case that doing continual learning by sequential Bayesian inference on the parameters of a neural network is very challenging in practice. I also think the ProtoCL method makes a useful albeit incremental contribution to the continual learning literature. Unfortunately, given the remaining issues I’m afraid that at present I cannot support acceptance of this paper.
> > > > > > > > > > > >
> > > > > > > > > > > > I’m not sure how flexible the decision timeline of TMLR is, but if possible I would be happy to allow for some more time to give the authors the opportunity to address these remaining issues. One option I would be happy with, and that would make me support acceptance of this paper, would be leaving out section 4 and 5 and adding a nuanced discussion that the Bayesian aspect of the proposed ProtoCL method might not actually contribute to the continual learning performance of this method (or alternatively the authors could provide evidence that the Bayesian aspect of their method does contribute).

---

> > > > > > > > > > > > > ### Author Response · Authors · 2023-02-16
> > > > > > > > > > > > > **Reply to official recommendation**
> > > > > > > > > > > > >
> > > > > > > > > > > > > Many thanks for the honest assessment. We are glad to hear that there are some valuable contributions to the continual learning community from our work. Also, regarding Sections 4 and 5, after multiple rounds of discussions, it seems like this has generated some interesting discussions which we believe the wider community could benefit from.
> > > > > > > > > > > > >
> > > > > > > > > > > > > However, to publish this work, we would be happy to remove Sections 4 and 5 and alter the abstract, introduction, and conclusion to reflect this. If this is the preferred option amongst reviewers. We will happily action this if there is consensus among all reviewers. We are hesitant to directly action this now as this is a change that hasn’t been reflected in the other reviewer’s preferences.

---

> > > > > > > > > > > > ### Author Response · Authors · 2023-02-16
> > > > > > > > > > > > **Response to remaining concerns**
> > > > > > > > > > > >
> > > > > > > > > > > > > OK, let’s focus on the point in time after which all data has been seen in the example of Figure 4B. (But in principle the same can be done for any point in time.) If I understand your claim correctly, you claim that at this point in time forgetting has taken place of task 1. But at the same time you agree that performance on both tasks would be exactly the same if the order of the data had been reversed (i.e., exactly the same data up to that point is seen but in a different order: now first task 2 and then task 1). To me this indicates that the lower performance on task 1 at this point in time cannot, in a meaningful way, be attributed to or described as forgetting.
> > > > > > > > > > > >
> > > > > > > > > > > > Thank you for clarifying. Indeed if the tasks are reversed and we first see data generated from the first task,  $N(1, 0.1^2)$ and then a smaller amount of data from the second task, $N(-1, 0.1^2)$ this will bias learning towards the first task. In this case there will be little learning of the second task. You are correct that this is not referred to as forgetting, but rather *intransigence* [1] i.e. the continual learning agent is resistant to learning new tasks. This behaviour is entirely consistent with other continual learning methods and experiments, e.g. [2] (Fig 1).
> > > > > > > > > > > >
> > > > > > > > > > > > [1] Chaudhry, Arslan, et al. "Riemannian walk for incremental learning: Understanding forgetting and intransigence." Proceedings of the European conference on computer vision (ECCV). 2018.
> > > > > > > > > > > >
> > > > > > > > > > > > [2] Goodfellow, Ian J., et al. "An empirical investigation of catastrophic forgetting in gradient-based neural networks." arXiv preprint arXiv:1312.6211 (2013).
> > > > > > > > > > > >
> > > > > > > > > > > > > Does this mean that the issue that you identify in section 4 and 5 is not addressed by the method you propose in section 7?
> > > > > > > > > > > >
> > > > > > > > > > > > In Sections 3, 4 and 5 we describe some of the limitations of working directly with sequential Bayesian inference in weight space, which are not immediately apparent from looking at Eq 5. In Section 7, with ProtoCL we claim that modelling the data-generating process is more fruitful and tackle the realistic and challenging problem of class-incremental learning. We do not tackle the problem of imbalanced task datasets with ProtoCL. Nor do we claim to do so. We claim that ProtoCL delivers good performance in class-incremental learning problems and provide evidence for this. However, this is an interesting direction for future work.
> > > > > > > > > > > >
> > > > > > > > > > > > > I do not think this is the case, unless you assume that task inference is flawless, which is an important assumption.
> > > > > > > > > > > >
> > > > > > > > > > > > You are correct, we are assuming that the correct latent variable $Z_t$ is inferred. Just like in a GMM or topic model for instance.
> > > > > > > > > > > >
> > > > > > > > > > > > > This was not my concern. My concern is that the Bayesian aspect of the method you propose (i.e. the sequential Bayesian inference of the parameters in the embedding space) does not contribute to the incremental learning at all.
> > > > > > > > > > > >
> > > > > > > > > > > > Our entire modelling approach takes a Bayesian perspective which is to place distributions over important parameters. We also maximize the posterior predictive to train our NN encoder weights and so do sequential Bayesian updates in our embedding space. So the ”Bayesian aspect” is not just the sequential Bayesian updates in the embedding space. We do not claim that the sequential Bayesian updates are the source of remembering. As noted in the paper if we remove the coreset of past data this will cause a loss in performance. We need to ensure that the encoder remembers how to map previous task data so that previous class Gaussian embeddings are remembered.

---

### Review · Reviewer_XPtH · 2023-01-17

**Summary Of Contributions:**

Authors explore sequential Bayesian inference for continual learning where it is usually believed that the performance degradation in a continual learning  setting known as catastrophic forgetting is mainly due to error accumulation in the process of posterior approximation. However, they show empirically that even by using a true BNN weight posterior and propagating it as a prior for new tasks (using a density estimator on Hamiltonian Monte Carlo samples) cannot prevent the forgetting issue in continual learning. They also provide a toy analytical example where exact inference is performed yet forgetting occurs due to so called model misspecification. Authors argue against using weight space sequential Bayesian inference and instead they propose to take a generative approach where they introduce a simple baseline called ProtoCL that can be thought as a probabilistic version of iCaRL (Rebuffi et al. 2017).



**Audience:**

Yes

**Claims And Evidence:**

No

**Requested Changes:**

Please see the weakness point #1 above. Authors should clarify if the reported results for VCL are accurate both in Figure 1 and Table 1. They seem to be lower than the originally reported results in the VCL paper as well as other works reporting VCL performance on Split MNIST and other datasets such as HAT by Serra et al 2018 or UCB by Ebrahimi et al. 2020

Serra, Joan, et al. "Overcoming catastrophic forgetting with hard attention to the task." International Conference on Machine Learning. PMLR, 2018.

Ebrahimi, Sayna, et al. "Uncertainty-guided Continual Learning with Bayesian Neural Networks." International Conference on Learning Representations. 2020.

**Strengths And Weaknesses:**

Strengths:

* This paper is mostly well-written and provides an interesting claim in BNNs where using true weight posteriors cannot mitigate forgetting.
* The paper attempts to provide empirical and analytical justifications to prove the claims.
* The paper claims that model misspecification can still cause forgetting despite using true posteriors. Solution to that is to specify a model by assessing the multi-task performance over all tasks a priori.

Weaknesses:
1. It appears that the experimental results provided in Fig 1 for VCL on Split MNIST dataset are different from what's reported in the original VCL paper (Figure 4) even for SGD results. Can authors confirm that this is a fair comparison? Claims made in this paper hugely depend on the accuracies reported here and I'd like to make sure the comparison is fair. Please note that the strengths mentioned above are only correct if the provided comparison is clarified to be correct.

2. One of the claims about the proposed baseline is its scalability yet the experiments of the paper are only done on MNIST, FMNIST, and CIFAR10 datasets which are far from real world and cannot generalize to datasets such as ImageNet. I encourage the authors to either reduce this claim or show its performance on a large scale dataset.

3. Abstract writing can be improved by earlier stating the main contributions of the main paper rather than leaving it to the last 2-3 sentences.

4. Some citation of prior works can be improved to be more accurate such as
i) In section 1, line 4, the citation for *catastrophic forgetting* is given as French 1995 but it dates back to older days of 1989 in the McCloskey & Cohen's paper.
ii) In the last two lines of section 4, the citation for online continual learning is given as De Lange et al., 2021 but I think the year in that citation should be 2019 (see the citation below). It was also discussed in papers by Aljundi in NeurIPS and CVPR of the same year.

* McCloskey, Michael, and Neal J. Cohen. "Catastrophic interference in connectionist networks: The sequential learning problem." Psychology of learning and motivation. Vol. 24. Academic Press, 1989. 109-165.

* De Lange, Matthias, et al. "Continual learning: A comparative study on how to defy forgetting in classification tasks." arXiv preprint arXiv:1909.08383 2.6 (2019): 2.

* Aljundi, Rahaf, et al. "Gradient based sample selection for online continual learning." Advances in neural information processing systems 32 (2019).

* Aljundi, Rahaf, Klaas Kelchtermans, and Tinne Tuytelaars. "Task-free continual learning." Proceedings of the IEEE/CVF Conference on Computer Vision and Pattern Recognition. 2019.

5. Some minor issues:
- Sec 2.1, line 8, between \hat{y} = g(x) and the next sentence there is a period missing.
- The order in which figure 2 and 3 are presented in paper can be changed such that figure 3 comes after figure 2.
- Section 5, line 3 (last line of page 5) --> did you mean "In" Section 3?

---

> ### Author Response · Authors · 2023-01-27
> **Author response to Reviewer XPtH (1/2)**
>
> Thank you for providing a thoughtful and constructive review of our manuscript.
>
> > One of the claims about the proposed baseline is its scalability yet the experiments of the paper are only done on MNIST, FMNIST, and CIFAR10 datasets which are far from real world and cannot generalize to datasets such as ImageNet. I encourage the authors to either reduce this claim or show its performance on a large scale dataset.
>
> Thanks for drawing this to our attention. Scalability has traditionally been more challenging for Bayesian deep learning approaches. We are not aware of any other Bayesian continual learning methods which scale beyond Split-CIFAR10 + 5 tasks of Split-CIFAR100 [1, 2]. Therefore by including experiments with 10 tasks of 10-way Split-CIFAR100 we believe that our work already demonstrates a step towards scalability with ProtoCL. We have edited the manuscript to clearly state that ProtoCL is competitive with other Bayesian CL methods in the abstract and we have added a sentence at the end of Section 7.
>
> [1] Pan, Pingbo, et al. "Continual deep learning by functional regularisation of memorable past." Advances in Neural Information Processing Systems 33 (2020): 4453-4464.
>
> [2] Rudner, Tim GJ, et al. "Continual Learning via Sequential Function-Space Variational Inference." International Conference on Machine Learning. PMLR, 2022.
>
> > Abstract writing can be improved by earlier stating the main contributions of the main paper rather than leaving it to the last 2-3 sentences.
>
> Thanks for the feedback we have re-written the abstract so that the contributions are clearly stated.
>
> > Some citation of prior works can be improved to be more accurate such as i) In section 1, line 4, the citation for catastrophic forgetting is given as French 1995 but it dates back to older days of 1989 in the McCloskey & Cohen's paper. ii) In the last two lines of section 4, the citation for online continual learning is given as De Lange et al., 2021 but I think the year in that citation should be 2019 (see the citation below). It was also discussed in papers by Aljundi in NeurIPS and CVPR of the same year.
>
> Many thanks for suggesting these more proper citations we have added these to the paper.
>
> Also, many thanks for pointing out minor issues and typos we have made these edits directly in the manuscript.

---

> > ### Author Response · Authors · 2023-01-27
> > **Author response to Reviewer XPtH (2/2)**
> >
> > > It appears that the experimental results provided in Fig 1 for VCL on Split MNIST dataset are different from what's reported in the original VCL paper (Figure 4) even for SGD results. Can authors confirm that this is a fair comparison? Claims made in this paper hugely depend on the accuracies reported here and I'd like to make sure the comparison is fair. Please note that the strengths mentioned above are only correct if the provided comparison is clarified to be correct.
> >
> > > Please see the weakness point #1 above. Authors should clarify if the reported results for VCL are accurate both in Figure 1 and Table 1. They seem to be lower than the originally reported results in the VCL paper as well as other works reporting VCL performance on Split MNIST and other datasets such as HAT by Serra et al 2018 or UCB by Ebrahimi et al. 2020
> >
> > SH VCL faithfully follows sequential Bayesian inference with a variational approximation. So in Fig 1 we explore results of Split-MNIST and compare SH VCL to SH SGD which doesn’t attempt sequential Bayesian inference. We observe that SH VCL barely prevents forgetting and barely performs better than SGD. We also include results for MH VCL as a comparison which performs much better, indicating that knowledge retention comes from parameter isolation. This observation is inline with the results from [2, 3, 4]. The VCL paper does not report results for SGD or SH VCL.
> >
> > We have used the original implementation from the authors of the VCL paper [1]. We use a BNN with identical hyperparameters on Split-MNIST as suggested by the authors, this is a fair comparison. Our results report slightly lower accuracies for MH VCL for tasks 1 through 3 than in the VCL paper [1], but otherwise the results are consistent with the literature [2, 3, 4]. When we use the implementation from [7] with the local reparameterization trick and longer training we improve results for MH VCL from $0.88 \pm 0.08$ to $0.96 \pm 0.02$, this could explain the difference in other papers.
> >
> > Other papers report good results which could be due to the use of larger networks of hidden size 1200 [5] (the paper you mention, [6], does not report results of VCL on Split-MNIST). Moreover, the primary takeaway from Fig 1 is that SH VCL performs poorly in comparison to MH VCL [2, 3, 4]. This throws into question the effectiveness of the variational approximation to the posterior in VCL. So motivates us to use HMC without requiring tricks for weight space sequential Bayesian inference such as multiple heads and coresets.
> >
> > The results in the Table 1 are for “class-incremental” learning where we ask the continual learning models to predict the exact class and do not give the task identifier during evaluation. This is the most realistic and practical continual learning scenario as we ask the agent to identify the individual classes seen so far. This is also the most difficult continual learning scenario. So this is a great way to test our approach to model the data-generating process, ProtoCL is designed specifically for class-incremental learning. We just want to reiterate that our aim is not to create a state-of-the-art Bayesian continual learning method. But to demonstrate that modelling the generative (“class-incremental”) continual learning process is a fruitful avenue for future continual learning models. The results reported in the Fig 1 (and in the VCL paper) are binary decision tasks were we ask the agent to classify between even and odd numbers, called “domain-incremental” learning. This is an easier continual learning problem and also valid for sequential Bayesian inference with a Bernoulli likelihood instead of a Categorical likelihood. This was a good starting point for attempting to perform sequential Bayesian inference using HMC. Also the 2-d synthetic datasets from previous work are also binary decision tasks, so we kept this setup inline with the literature.
> >
> > [1] Nguyen, Cuong V., et al. "Variational Continual Learning." International Conference on Learning Representations. 2018.
> >
> > [2] Rudner, Tim GJ, et al. "Continual Learning via Sequential Function-Space Variational Inference." International Conference on Machine Learning. PMLR, 2022.
> >
> > [3] Farquhar, Sebastian, and Yarin Gal. "Towards robust evaluations of continual learning." arXiv preprint arXiv:1805.09733 (2018).
> >
> > [4] Farquhar, Sebastian, and Yarin Gal. "A unifying bayesian view of continual learning." arXiv preprint arXiv:1902.06494 (2019).
> >
> > [5] Ebrahimi, Sayna, et al. "Uncertainty-guided Continual Learning with Bayesian Neural Networks." International Conference on Learning Representations. 2020.
> >
> > [6] Serra, Joan, et al. "Overcoming catastrophic forgetting with hard attention to the task." International Conference on Machine Learning. PMLR, 2018.
> >
> > [7] Swaroop, Siddharth, et al. "Improving and understanding variational continual learning." arXiv preprint arXiv:1905.02099 (2019).

---

### Review · Reviewer_LRZY · 2023-01-18

**Summary Of Contributions:**

The paper first answers the question of whether having access to the true posterior is guaranteed to prevent catastrophic forgetting in Bayesian neural networks. It experiments with the gold-standard Bayesian inference method HMC (augmented with an extra density estimator) on Split-MNIST and finds that the catastrophic forgetting issue remains. The paper then studies simple analytical examples of sequential Bayesian inference and highlights the issue of model misspecification which can lead to sub-optimal continual learning performance despite exact inference. It also discusses how task data imbalances can cause forgetting. The authors, at last, derive a new algorithm called Prototypical Bayesian Continual Learning that can be thought of as a probabilistic version of iCaRL (Rebuffi et al. 2017), and find that it can beat the considered baselines.

**Audience:**

Yes

**Claims And Evidence:**

No

**Requested Changes:**

The authors should provide convincing clarification to the raised question in the Weaknesses part.

**Strengths And Weaknesses:**

# Strengths
The paper indeed provides new insights and is likely to help to correct the notion that non-exact Bayesian inference causes error accumulation and hence performance degradation in Bayesian continual learning. If the arguments made in the paper can be fully confirmed, I think the paper can form a valuable and prompt contribution to both Bayesian learning and continual learning communities.

The argument that model misspecification causes the failure of Bayesian continual learning is interesting. Model misspecification is also an increasingly important topic in the Bayesian inference community.

# Weaknesses
- In general, I am not satisfied with the writing and presentation of this paper. There are issues with the figures and references. There are also multiple typos.

- The conclusions drawn from the results in Section 2 and 3 are questionable. On one hand, the authors concern only with the domain-incremental CL, so the scope of the conclusions should be pointed out; on the other hand, the MNIST dataset is less convincing and it is hard to distinguish if the mentioned issue is general.

- Technically, I don't think the GMM estimator on the HMC samples is a good modeling choice. As the authors agreed, "the GMM removes probability mass over areas of the BNN weight space posterior which is important for the new task". Therefore, is there a probability that it is your technical limitation that makes you wrongly conclude that exact Bayesian inference cannot cause gains for Bayesian CL? You said you have tried RealNVP and it failed. But I think there are better Normalizing Flows now that can be easily adopted. You can also resort to non-parametric estimators (e.g., Nonparametric Score Estimators, ICML 2020) to avoid introducing a parametric density estimator that has inherent limitations. I really expect to see the new results of using these new techniques and worry that the conclusions may be different.

- The discussion on model misspecification in Section 4 is not convincing. As we know, in practice, we use deep neural networks (as you have done in the MNIST case) to parameterize the likelihood instead of a point parameter. In that case (i.e., using DNNs), is there still model misspecification? I think a study that can resolve this question is that you use a known model (e.g., a two-layer MLP) to generate a dataset and perturb the data with Gaussian noise, then you set up a model in the same family and check if Bayesian CL can works.

- Is iCaRL essentially a online EM model, where the prior follows a Dirichlet process?

---

> ### Author Response · Authors · 2023-01-27
> **Author response to Reviewer LRZY (1/2)**
>
> We would like to thank you for reviewing our paper and for your constructive feedback.
>
> > In general, I am not satisfied with the writing and presentation of this paper. There are issues with the figures and references. There are also multiple typos.
>
> We have fixed multiple typos, added references, and updated figures according to feedback from the reviews.
>
> > The conclusions drawn from the results in Section 2 and 3 are questionable. On one hand, the authors concern only with the domain-incremental CL, so the scope of the conclusions should be pointed out; on the other hand, the MNIST dataset is less convincing and it is hard to distinguish if the mentioned issue is general.
>
> Our primary concern is performing sequential Bayesian inference. In the continual learning literature there are different continual learning scenarios: “task-incremental”, “domain-incremental” and “class-incremental” [1]. The “task-incremental” learning scenario involves access to the task identifier to aid predictions, this is not a requirement from sequential Bayesian inference so we do not consider it. In “domain-incremental” and “class-incremental” scenarios both do not have task information when making predictions and consider Bernoulli and Categorical likelihoods respectively. So both of these are compatible with sequential Bayesian inference. It is generally known within the continual learning literature that domain incremental learning with NNs is easier [1], so this is a good scenario to attempt initially with HMC. We consider two synthetic datasets from [2] and [3] where the data is drawn from a 2-d input space. Both of these datasets are ‘“domain-incremental”.
>
> In Fig 1 we show a domain-incremental or SH VCL example with Split-MNIST as this compatible with sequential Bayesian inference. We also include (SH) SGD as a reference in addition to MH VCL to show how task heads are very effective again as a reference. These results have also been observed for other datasets, for instance Split-FashionMNIST and Split-CIFAR10 [1, 4]. So this issue is not particular to Split-MNIST but a well establish result in the community.
>
> [1] van de Ven, Gido M., Tinne Tuytelaars, and Andreas S. Tolias. "Three types of incremental learning." Nature Machine Intelligence (2022): 1-13.
>
> [2] Pan, Pingbo, et al. "Continual deep learning by functional regularisation of memorable past." Advances in Neural Information Processing Systems 33 (2020): 4453-4464.
>
> [3] Henning, Christian, et al. "Posterior meta-replay for continual learning." Advances in Neural Information Processing Systems 34 (2021): 14135-14149.
>
> [4] Farquhar, Sebastian, and Yarin Gal. "Towards robust evaluations of continual learning." arXiv preprint arXiv:1805.09733 (2018).
>
> > Technically, I don't think the GMM estimator on the HMC samples is a good modeling choice. As the authors agreed, "the GMM removes probability mass over areas of the BNN weight space posterior which is important for the new task". Therefore, is there a probability that it is your technical limitation that makes you wrongly conclude that exact Bayesian inference cannot cause gains for Bayesian CL? You said you have tried RealNVP and it failed. But I think there are better Normalizing Flows now that can be easily adopted. You can also resort to non-parametric estimators (e.g., Nonparametric Score Estimators, ICML 2020) to avoid introducing a parametric density estimator that has inherent limitations. I really expect to see the new results of using these new techniques and worry that the conclusions may be different.
>
> Thanks for your comment, however we found that the GMM to be a good density estimator for our continual learning problems in Section 3. Initially we were concerned that the GMM would be too restrictive however we found that the GMM captures the posterior distribution in the sense that when we remove the HMC samples and sample from the GMM density estimator we get perfect performance on the current task from the synthetic 2-d datasets.
>
> Many thanks for suggesting further work in the NP literature for us to look into. RealNVP is a robust normalizing flow baseline for density estimation and is cheap to evaluate log probabilities so doesn’t slow down HMC sampling. Despite this, we observed some numerical errors on certain runs which shows just how difficult it is to perform density estimation in complex neural network weight space. Other better NF methods like FFJORD [1] are very expensive to evaluate log probabilities so make the HMC sampling process prohibitively expensive. So taking this into account our conclusion that sequential Bayesian inference over NN weights is very difficult is correct. We are not saying that exact Bayesian inference doesn’t work, it does according to Eq. 5 with a BNN and considering the datasets in Section 3.
>
> [1] Grathwohl, Will, et al. "Ffjord: Free-form continuous dynamics for scalable reversible generative models." arXiv preprint arXiv:1810.01367 (2018).

---

> > ### Author Response · Authors · 2023-01-27
> > **Author response to Reviewer LRZY (2/2)**
> >
> > > The discussion on model misspecification in Section 4 is not convincing. As we know, in practice, we use deep neural networks (as you have done in the MNIST case) to parameterize the likelihood instead of a point parameter. In that case (i.e., using DNNs), is there still model misspecification? I think a study that can resolve this question is that you use a known model (e.g., a two-layer MLP) to generate a dataset and perturb the data with Gaussian noise, then you set up a model in the same family and check if Bayesian CL can works.
> >
> > To test for model-mispecification we ensure that our model is able to achieve good performance when trained on the union of all the task datasets. So, for our synthetic dataset in Fig 3 we use a 2 layer network with width of size 10. This network is able to achieve a multi-task accuracy of 1.0 with HMC. So we know that our posterior $p(\theta | D_1, \ldots D_5)$ is a good one and the model, which is a DNN, is not misspecified. So sequential Bayesian inference can recover this posterior, Eq 5. The same argument can be applied to Split-MNIST.
> >
> > We could generate task datasets from 2 layer BNNs, we could then assess whether a single BNN is able to obtain good performance on the union of all task datasets. If so, then our model is well specified. Then we can use HMC with density estimators like in our manuscript to perform sequential Bayesian inference. This is the same setup as the synthetic datasets from the CL literature we study in Section 3 and so this does not offer any new insights which are not already in the manuscript.
> >
> > > Is iCaRL essentially a online EM model, where the prior follows a Dirichlet process?
> >
> > iCaRL is a continual learning method that constructs class prototypes and classifies points according to how close a data point embedding is to the class prototype embedding. This is similar to ProtoCL, in that ProtoCL classifies a point according to the probability it was generated by a Gaussian class embedding. iCaRL also introduces a distillation loss and coreset construction heuristics to help it maintain previous task / class prototypes. iCaRL doesn’t use an EM to maximize a likelihood nor does it use a Dirichlet distribution to track class probabilities.

---

### Decision · Action_Editors · 2023-03-25

**Recommendation:** Reject

**Comment:**

The paper received two reject recommendations and two weak accept recommendations.

While the paper and the insights from the discussion with the reviewers, might be valuable to the community, unfortunately, I believe the paper requires a major revision prior to acceptance.

I see two potential paths, but there might be others, the authors might want to consider:
1. In the follow-up discussion, one of the reviewers recommending "weak accept" agreed with one of the reject reviewers, that "leaving out sections 4 and 5 and adding a nuanced discussion that the Bayesian aspect of the proposed ProtoCL method might not actually contribute to the continual learning performance of this method". Also, the authors offered to remove sections 4 and 5. While this seems to be a possible direction, this at the same time removes a significant portion and insight of the paper. Additionally, the mentioned "nuanced discussion" will require a careful review as this is a major part of the paper (especially if also sections 4 and 5 are removed). So while this seems to address the reviewer's concerns, I am not sure the paper can stand on its own in this case and will require another complete review.

2. Another option might be to significantly strengthen section 7, by analyzing ProtoCL in relation to the settings/discussions from sections 4 and 5 as well as evaluating the Bayesian aspect of ProtoCL.

The path for major revision in TMLR is Reject and a Resubmission, see https://jmlr.org/tmlr/editorial-policies.html
> Authors of a rejected submission may revise and resubmit their paper, but it will need to be entered as a new submission and a link provided to the previously rejected submission as well as a description of the changes made since.


Overall if the paper is carefully revised and reviewed I think this could become an insightful work.

**Audience:**

I think the paper could be of interest in the continual learning community if all aspects would be extensively evaluated for the final approach (See 2. in the next question); in contrast, removing section 4 and 5, seem to limit the potential interest.

**Claims And Evidence:**

The individual aspects of the paper are reasonably well supported, including the likely strongest point of the paper:
The proposed ProtoCL approach and the evaluation of several incremental learning datasets/tasks.

However, the claims of the paper are a bit disconnected and potentially misleading:
1. The initial analysis (model misspecification and imbalanced task data, in Sections 4 and 5) is disconnected from the later parts of the approach and main experimental evaluation.
2. The analysis w.r.t. data imbalance seems to be less of a continual learning issue rather than a data imbalance problem.

Additionally, most of the experiments are on very simplistic data, such as MNIST, questioning the generalizability of the conclusions